# Depletion of mitochondrial methionine adenosyltransferase α1 triggers mitochondrial dysfunction in alcohol-associated liver disease

Lucía Barbier-Torres [1], Ben Murray[1], Jin Won Yang [1,2], Jiaohong Wang[1], Michitaka Matsuda[1], Aaron Robinson[3], Aleksandra Binek[3], Wei Fan[1], David Fernández-Ramos [4], Fernando Lopitz-Otsoa [4], Maria Luque-Urbano[4], Oscar Millet[4], Nirmala Mavila[1], Hui Peng[1], Komal Ramani [1], Roberta Gottlieb [3], Zhaoli Sun[5], Suthat Liangpunsakul[6,7,8], Ekihiro Seki [1], Jennifer E. Van Eyk [3], Jose M. Mato [4] & Shelly C. Lu [1✉]

MATα1 catalyzes the synthesis of S-adenosylmethionine, the principal biological methyl donor. Lower MATα1 activity and mitochondrial dysfunction occur in alcohol-associated liver disease. Besides cytosol and nucleus, MATα1 also targets the mitochondria of hepatocytes to regulate their function. Here, we show that mitochondrial MATα1 is selectively depleted in alcohol-associated liver disease through a mechanism that involves the isomerase PIN1 and the kinase CK2. Alcohol activates CK2, which phosphorylates MATα1 at Ser114 facilitating interaction with PIN1, thereby inhibiting its mitochondrial localization. Blocking PIN1-MATα1 interaction increased mitochondrial MATα1 levels and protected against alcohol-induced mitochondrial dysfunction and fat accumulation. Normally, MATα1 interacts with mitochondrial proteins involved in TCA cycle, oxidative phosphorylation, and fatty acid β-oxidation. Preserving mitochondrial MATα1 content correlates with higher methylation and expression of mitochondrial proteins. Our study demonstrates a role of CK2 and PIN1 in reducing mitochondrial MATα1 content leading to mitochondrial dysfunction in alcohol-associated liver disease.

[1] Karsh Division of Gastroenterology and Hepatology, Cedars-Sinai Medical Center, Los Angeles, CA 90048, USA. [2] College of Pharmacy, Woosuk University, Wanju, South Korea. [3] Smidt Heart Institute, Cedars-Sinai Medical Center, Los Angeles, CA 90048, USA. [4] Precision Medicine and Metabolism, CIC bioGUNE, BRTA, CIBERehd, Technology Park of Bizkaia, 48160 Derio, Bizkaia, Spain. [5] Department of Surgery and Transplant Biology Research Center, Johns Hopkins University School of Medicine, Baltimore, MD, USA. [6] Division of Gastroenterology and Hepatology, Department of Medicine, Indiana University School of Medicine, Indianapolis, IN, USA. [7] Department of Biochemistry and Molecular Biology, Indiana University School of Medicine, Indianapolis, IN, USA. [8] Roudebush Veterans Administration Medical Center, Indianapolis, IN, USA. ✉email: shelly.lu@cshs.org

Alcohol-associated liver disease (ALD) is a leading cause of liver injury resulting from chronic and abusive alcohol consumption. The spectrum of ALD ranges from alcoholic steatosis, to hepatitis, fibrosis, cirrhosis, and hepatocellular carcinoma (HCC)[1]. Cirrhosis causes 1.16 million deaths worldwide and over 50% are due to alcohol abuse[2]. Despite major advances made in identifying potential pathways and therapeutic targets, there is no effective therapy besides abstinence. Uncovering novel molecular mechanisms of ALD pathogenesis that can lead to new treatments are urgently needed.

Methionine adenosyltransferase α1 (MATα1) catalyzes the synthesis of S-adenosylmethionine (SAMe), the principal methyl donor and precursor of the antioxidant glutathione (GSH), in the liver[3,4]. MATα1 is highly expressed in normal liver but its activity and SAMe levels are reduced in disease including ALD[5]. MATα1 was identified in the cytosol where it produces SAMe and we reported its presence in the nucleus where it acts as a transcription co-factor[6]. Recently we showed that MATα1 is also localized in the mitochondria of hepatocytes to regulate their function[3]. The role of mitochondrial MATα1 in ALD, where mitochondrial injury and dysfunction are major and early events[7], has not been explored.

Peptidyl-prolyl cis/trans isomerase NIMA-interacting 1 (PIN1) is a highly conserved enzyme that isomerizes phosphorylated serine/threonine-proline motifs in certain proteins[8]. By promoting the cis–trans-isomerization, PIN1 can alter the conformation, catalytic activity, interactions, localization, and stability of its targets[8]. PIN1 can promote or inhibit the mitochondrial translocation of target proteins thereby regulating mitochondrial function[9], oxidative stress[9,10], and apoptosis[11,12]. PIN1 is overexpressed in numerous cancers and regulates processes like cell metabolism, mobility, proliferation, and survival[13]. In the liver, PIN1 is implicated in fibrosis[14], non-alcoholic steatohepatitis (NASH)[15], and carcinogenesis[16], but its role in ALD has never been investigated. MATα1 contains a PIN1 binding motif (Ser114-Pro115), which led us to hypothesize that MATα1 could be a PIN1 target.

Casein kinase 2 (CK2) is a constitutively active serine-threonine kinase that phosphorylates hundreds of substrates and controls multiple signaling pathways[17,18]. CK2 has been implicated in many human diseases including NASH[19] and HCC[20] and was found upregulated in ethanol-treated hepatocytes[21].

In this study, we discovered that in liver, alcohol impairs MATα1 mitochondrial targeting by inducing CK2-mediated phosphorylation of Ser114 and binding to PIN1. We demonstrated that MATα1 normally interacts with mitochondrial proteins that participate in important mitochondrial metabolic pathways and that preserving mitochondrial MATα1 correlates with higher methylation and expression of mitochondrial proteins. By using a MATα1 mutant that cannot bind to PIN1, we observed that mitochondrial MATα1 protects against ethanol-induced mitochondrial injury. Silencing Csnk2a1 and Pin1 also preserved MATα1 mitochondrial content and protected against ethanol-induced mitochondrial dysfunction. Collectively, our study reveals that PIN1 negatively regulates MATα1 mitochondrial localization, and that alcohol enhances PIN1-MATα1 interaction resulting in selective depletion of mitochondrial MATα1, which may play an important role in ALD pathogenesis.

## Results

**Mitochondrial MATα1 is selectively reduced in ALD.** To investigate whether an alteration in mitochondrial MATα1 could be involved in the pathogenesis of ALD, we evaluated MATα1 levels in different ALD models. Liver samples from normal and alcoholic hepatitis (AH) patients revealed a 40% reduction in total MATα1 protein levels (Fig. 1a, b) and a 70% reduction in MAT1A mRNA levels (Fig. S1a), the latter being consistent with the GSE28619 database (Fig. S1b). Interestingly, whilst cytosolic MATα1 levels were unchanged (Fig. 1a, b), mitochondrial MATα1 content was reduced by almost 80% in AH livers compared to controls (Fig. 1a, b). Similarly, livers of mice subjected to the NIAAA model which consists of a 10-day alcohol feeding followed by a single binge[22] showed 30% reduction in total MATα1 protein levels (Fig. 1c, d). In this model, steatosis and inflammation occurred in the liver (Fig. S1c) and Mat1a mRNA levels were upregulated (Fig. S1d). Similar to human livers, cytosolic MATα1 content was unchanged but mitochondrial MATα1 content was 85% lower in the ethanol group (Fig. 1c, d). Last, the mouse hepatocyte cell line alpha mouse liver 12 (AML-12) treated with ethanol (100 mM for 48 h) recapitulated human AH and in vivo findings. Ethanol reduced MATα1 total protein levels by 30% (Fig. 1e) and mitochondrial MATα1 content by 85% (Fig. 1f, g). Mat1a mRNA levels were also increased in this model (Fig. S1e). The purity of the fractions was determined using specific markers (Fig. S1f, g and Fig. 1f). Altogether, these findings demonstrate that alcohol reduces selectively mitochondrial MATα1 content. Although PIN1 expression was previously shown to be stimulated by alcohol in cardiomyocytes[23], its expression was unchanged in ALD (Fig. 1 and Fig. S1).

**Alcohol enhances PIN1 and MATα1 interaction in the liver.** Ser114-Pro115 is the only potential PIN1 binding site in MATα1 and is conserved between human and mouse (Fig. 2a). To see if MATα1 could be a PIN1 target we first evaluated their interaction. Co-immunoprecipitation (IP) analyses in human (Fig. 2b) and mouse hepatocytes (Fig. 2c) confirmed that the two proteins interact. Further IP analyses using MATα1 and PIN1 recombinant proteins and antibody-conjugated beads confirmed that MATα1 and PIN1 are direct binding partners (Fig. 2d).

Next, we evaluated MATα1 and PIN1 interaction in ALD. Figure 2e shows the interaction between PIN1 and MATα1 in human alcoholic cirrhotic livers is 3.3-fold that of normal human livers. Similarly, ethanol-fed mouse livers showed PIN1-MATα1 interaction is 2-fold that of pair-fed mice (Fig. 2f, g). PIN1 is not present in hepatic mitochondria (Fig. S1f, g) and we found that its interaction with MATα1 takes place in the cytosol (Fig. 2h). Last, in AML-12 cells treated with ethanol, the interaction between the two proteins is 3.5-folds of control (Fig. 2i and j). Altogether, these results confirm that MATα1 is a PIN1 target, and that alcohol enhances their interaction.

**PIN1 negatively regulates MATα1 mitochondrial targeting.** PIN1 may alter the stability and expression level of target proteins by inducing cis–trans isomerization of specific serine/threonine-proline bonds[8]. To study whether PIN1 regulates MATα1 expression, PIN1 levels were silenced or overexpressed in AML-12 and HepG2 cells, and MATα1 protein levels were evaluated. Although PIN1 did not alter MATα1 protein levels (Fig. 3a, b), it negatively regulated its mitochondrial content in hepatocytes. Whilst PIN1 knockdown increased by 300 and 150% the mitochondrial content of MATα1 in AML-12 and HepG2 cells, respectively (Fig. 3c, e), PIN1 overexpression reduced MATα1 mitochondrial content by 60% in AML-12 and 40% in HepG2 cells (Fig. 3d, f). To determine whether PIN1 is involved in MATα1 mitochondrial depletion in ALD, we silenced Pin1 in AML-12 cells treated with ethanol. As shown in Fig. 3g, h, both total and mitochondrial MATα1 levels were preserved in the absence of PIN1, suggesting that PIN1 plays a key role in reducing MATα1 content in ALD. Along with increased

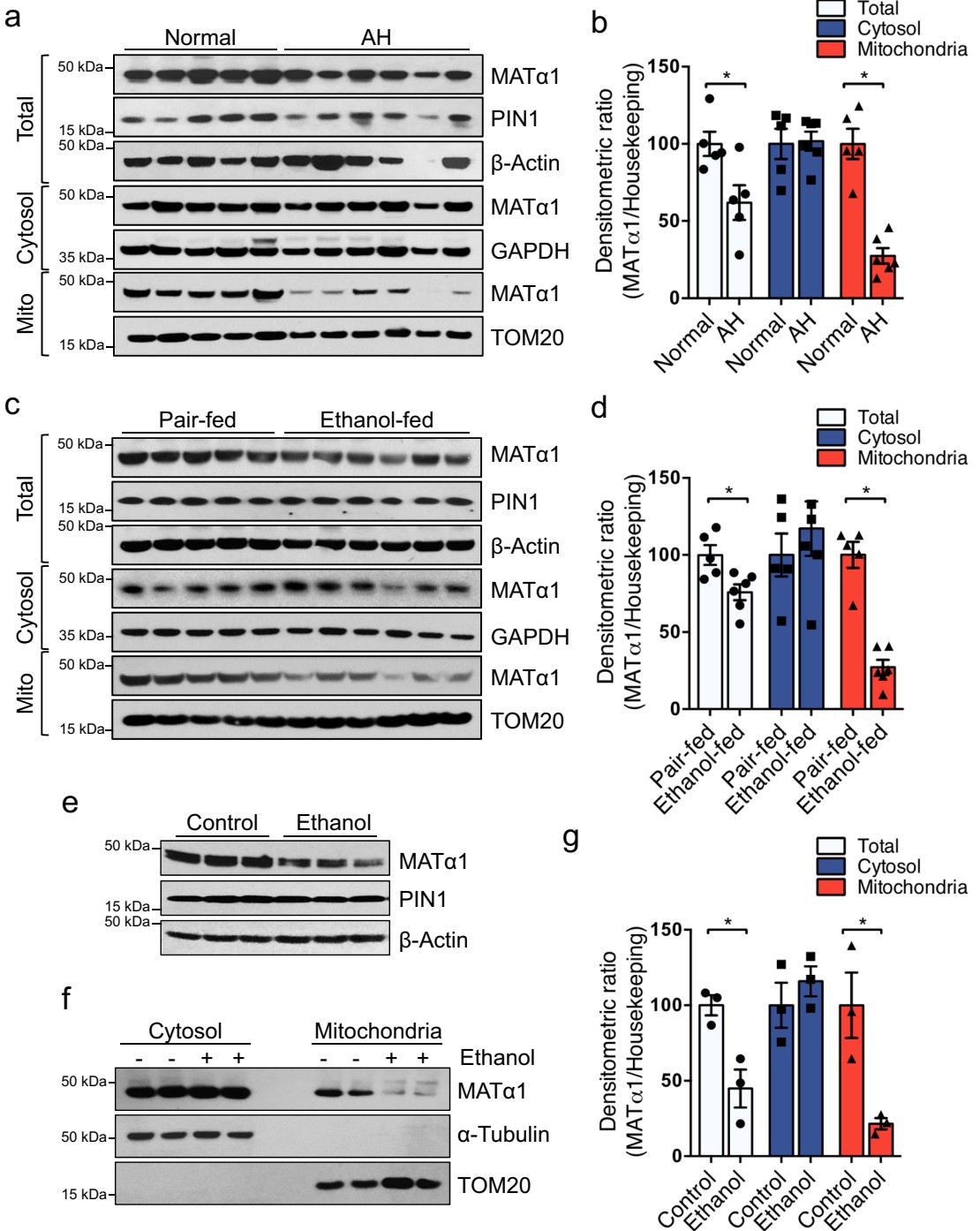

**Fig. 1 Mitochondrial MATα1 level is selectively reduced in alcohol-associated liver disease.** Western blots (**a**) and densitometry analyses (**b**) of total, cytosolic and mitochondrial MATα1 in human normal and AH livers ($n = 5$ normal and $n = 6$ AH independent patients. $p = 0.027$ for total and $p = 0.0006$ for mitochondrial MATα1). Western blots (**c**) and densitometry analyses (**d**) of total, cytosolic and mitochondrial MATα1 in pair-fed and ethanol-fed mouse livers ($n = 5$ pair-fed and n = 6 ethanol-fed independent animals. $p = 0.049$ for total and $p = 0.0004$ for mitochondrial MATα1). Western blots (**e**, **f**) and densitometry analyses (**g**) of total, cytosolic and mitochondrial MATα1 in AML-12 cells treated with ethanol ($n = 3$ independent experiments. $p = 0.018$ for total and $p = 0.024$ for mitochondrial MATα1). Data are presented as mean values ± SEM. *$p < 0.05$. Statistical significance was determined by using two-tailed, one-sample $t$-test. Source data are provided as a Source data file. Total lysate is shown in white, cytosolic fraction in blue and mitochondrial fraction in red. AH alcoholic hepatitis.

mitochondrial MATα1, *Pin1* silencing in AML-12 cells significantly increased ATP (Fig. 3i) and reduced triglycerides levels both at baseline and after ethanol treatment (Fig. 3j, k). Finally, to

confirm that PIN1 regulates MATα1 mitochondrial localization through isomerization, we overexpressed PIN1 wild-type (WT) and the catalytic mutant R68A/R69A and evaluated total and

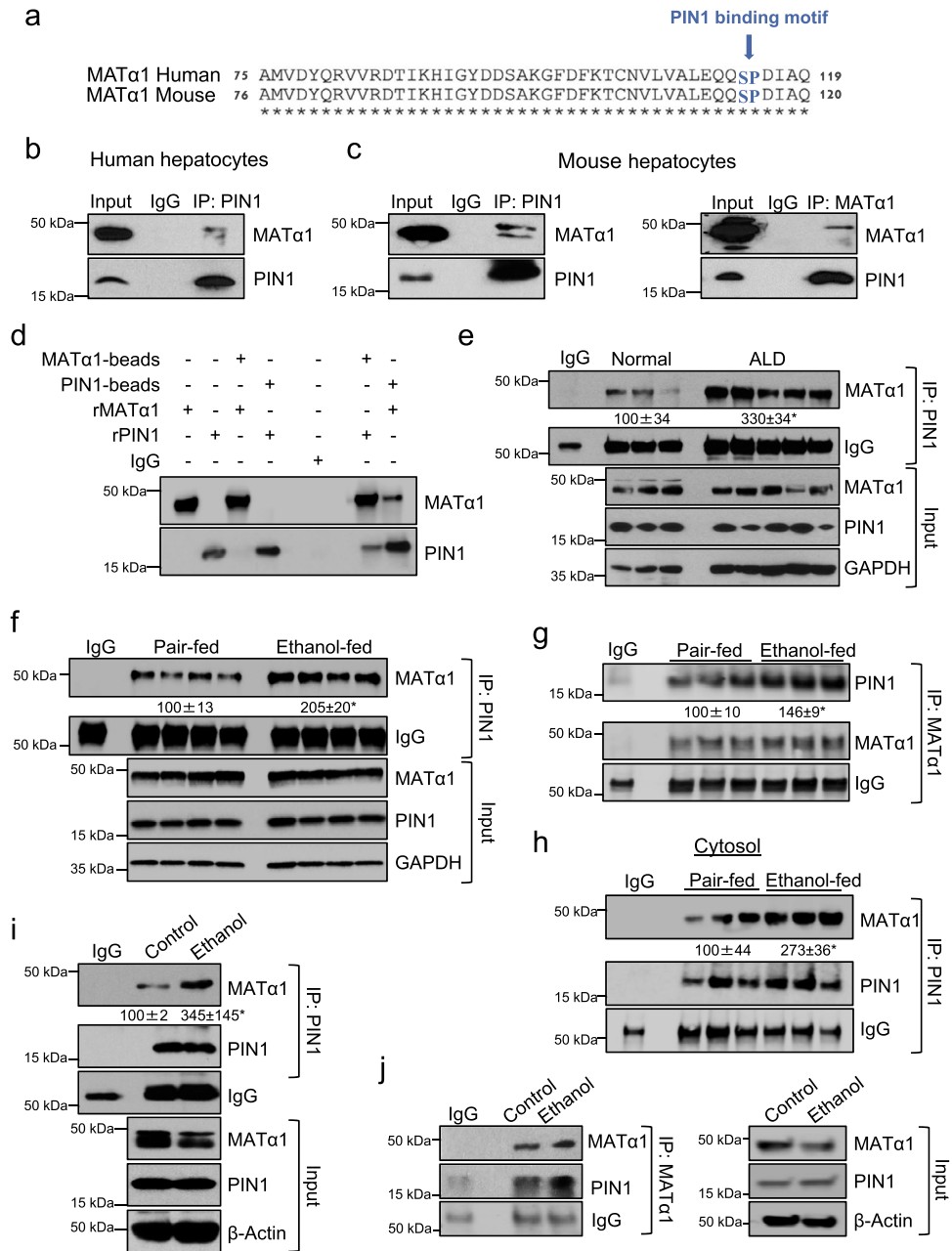

**Fig. 2 Alcohol enhances PIN1 and MATα1 interaction in the liver. a** Human and mouse MATα1 protein sequences showing that PIN1 binding motif is conserved between species. **b** IP analysis of human hepatocytes lysate using anti-PIN1 antibody followed by western blot analysis against MATα1. **c** IP analysis of WT mouse liver hepatocytes lysate with anti-PIN1 (left panel) or anti-MATα1 (right panel) antibody followed by western blot analysis against MATα1 (left panel) or PIN1 (right panel). **d** IP analysis using PIN1 and MATα1 recombinant proteins and beads conjugated to anti-PIN1 and anti-MATα1 antibodies. IgG was used as a negative control. IP analysis of human normal and end-stage cirrhotic ALD liver lysates (**e**) ($n = 3$ normal and $n = 5$ ALD independent patients. $p = 0.044$), pair-fed and ethanol-fed mouse livers whole lysates (**f**, $n = 4$ pair-fed and $n = 4$ ethanol-fed independent animals. $p = 0.002$) and (**g**, $n = 3$ pair-fed and $n = 3$ ethanol-fed independent animals $p = 0.027$) and cytosolic fractions (**h**, $n = 3$ pair-fed and $n = 3$ ethanol-fed independent animals. $p = 0.038$), and AML-12 cells treated with ethanol (**i** and **j**, $n = 3$ independent experiments. $p = 0.005$) using anti-PIN1 or anti-MATα1 antibodies followed by western blot analysis against MATα1 or PIN1. Densitometry analysis are shown; ethanol-fed vs pair-fed; and ethanol vs control. Results are shown as mean ± SEM. *$p < 0.05$. Statistical significance was determined by using two-tailed, one-sample $t$-tests. Source data are provided as a Source data file. ALD alcohol-associated liver disease, IP immunoprecipitation, r recombinant.

mitochondrial MATα1 levels in HepG2 cells. R68/R69 are in the peptidyl-prolyl isomerase flexible loop of PIN1 and their mutation reduces isomerase activity by >500-fold[24]. As shown in Fig. 3L, unlike WT PIN1, inactive PIN1 did not lower mitochondrial MATα1.

PIN1 can regulate proteasomal degradation of target proteins, but we found that this is not the mechanism by which alcohol reduces

MATα1. In fact, ethanol inhibited proteasomal activity, as seen by the accumulation of ubiquitinated proteins, and proteasome inhibition with MG132 did not prevent ethanol-induced lowering of MATα1 levels (Fig. S2a). However, we found that ethanol activates autophagy in AML-12 cells (Fig. S2b and c), and autophagy inhibition with chloroquine prevented ethanol-induced MATα1 depletion (Fig. S2b), confirming that ethanol promotes MATα1 lysosomal degradation.

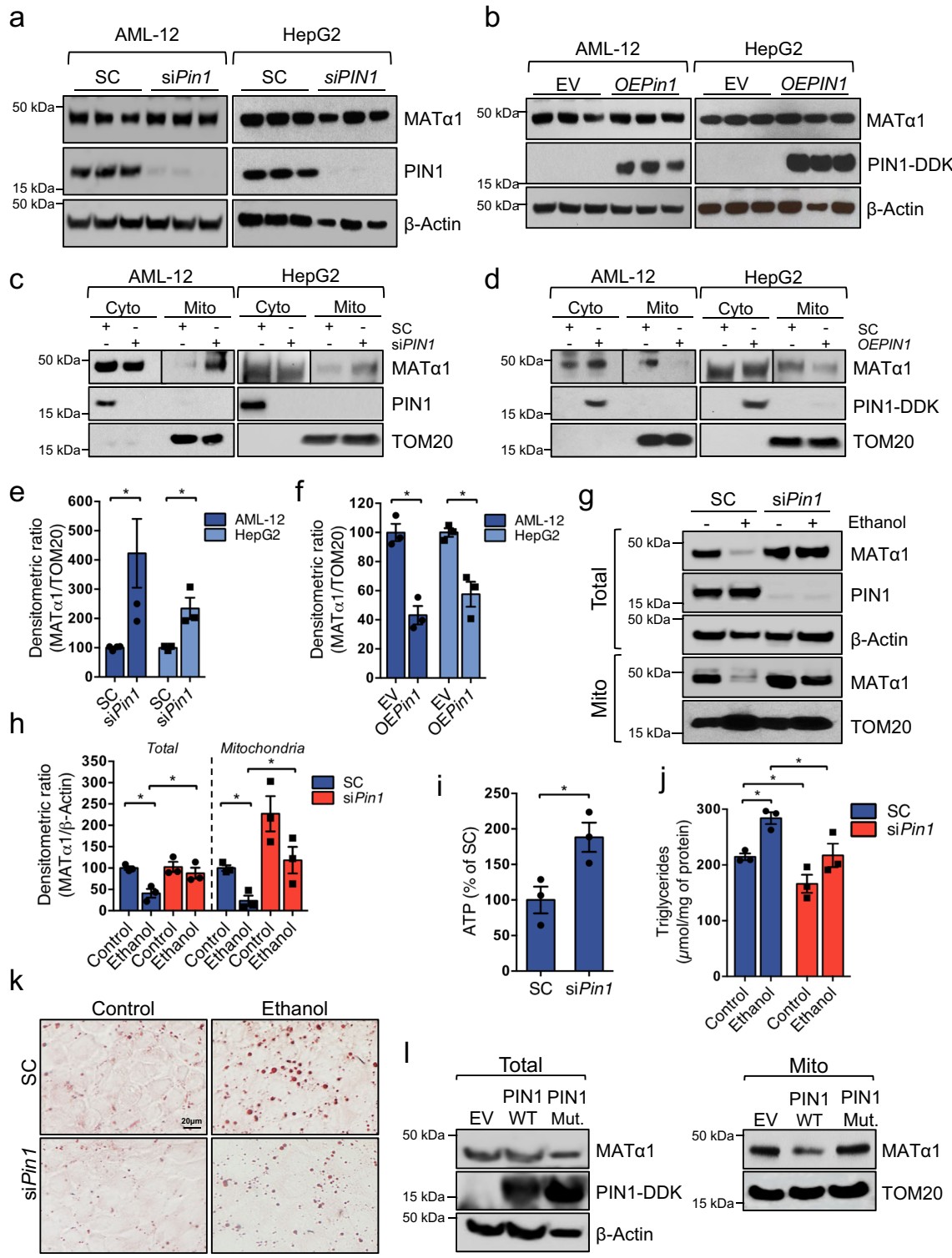

**Alcohol-induced MATα1 phosphorylation at Ser114 is required for PIN1 binding**. As previously mentioned, Ser114-Pro115 is the only consensus PIN1 binding site in MATα1. Since PIN1 recognizes and isomerizes phosphorylated serine/threonine-proline motifs in its targets[8], Ser114 must be phosphorylated for PIN1 to bind to MATα1. To assess MATα1 phosphorylation at serine residues in ALD we performed co-IP analyses using a phosphoserine antibody in human AH samples, NIAAA model mouse livers, and ethanol-treated AML-12 cells. Western blots after co-IP showed MATα1 serine phosphorylation increased by

250% in AH livers, 170% in ethanol-fed mouse livers, and 140% in AML-12 cells treated with ethanol (Fig. 4a). Using Phos-Tag[TM] gels, which capture phosphorylated Ser/Thr/Tyr and His/Asp/Lys, and cytosolic and mitochondrial fractions of ethanol-treated AML-12 cells, we found that ethanol promotes MATα1 phosphorylation in the cytosol (Fig. 4b). To examine MATα1 Ser114 phosphorylation, MATα1 co-IP followed by mass spectrometry (MS) proteomics, which detects mainly phosphorylated serine and threonine, were performed. Proteomic analyses confirmed phosphorylation of MATα1 at Ser114 in mouse livers (Figs. 4c

**Fig. 3 PIN1 negatively regulates MATα1 mitochondrial targeting.** Western blots of MATα1 and PIN1 in AML-12 and HepG2 cells after *PIN1* silencing (**a**) and overexpression (**b**). Western blots and densitometry analyses of cytosolic and mitochondrial MATα1 and PIN1 in AML-12 and HepG2 cells after *PIN1* silencing (**c** and **e**, AML-12: $n = 4$ independent experiments, $p = 0.033$ and HepG2: $n = 3$ independent experiments, $p = 0.022$) and overexpression (**d** and **f**, AML-12: $n = 3$ independent experiments, $p = 0.003$ and HepG2: $n = 3$ independent experiments, $p = 0.009$). AML-12 are shown in dark blue and HepG2 in light blue. **g** Western blot and **h** densitometry analyses of total and mitochondrial MATα1 in AML-12 cells after *PIN1* silencing and ethanol treatment ($n = 3$ independent experiments, $p = 0.006$ for total and $p = 0.005$ for mitochondrial MATα1 SC ethanol vs SC control; $p = 0.047$ for total and $p = 0.046$ for mitochondrial MATα1 siPin1 ethanol vs SC ethanol). SC is shown in blue and siPin1 in red. **i** ATP levels in AML-12 cells after *Pin1* silencing ($n = 3$ independent experiments, $p = 0.034$). **j** Triglycerides levels ($n = 3$ independent experiments, $p = 0.004$ SC ethanol vs control; $p = 0.048$ siPin1 control vs SC control; and $p = 0.049$ siPin1 ethanol vs SC ethanol) and **k** Oil red O staining in AML-12 after *Pin1* silencing and ethanol treatment. SC is shown in blue and siPin1 in red. **l** Western blot of total and mitochondrial MATα1 in HepG2 cells after PIN1 WT or R68A/R69A catalytic mutant overexpression. *$p < 0.05$. Statistical significance was determined by using two-tailed, one-sample *t*-test for treatment comparisons and ANOVA test for group comparisons. Results are shown as mean ± SEM. Source data are provided as a Source data file. DDK DYKDDDDK-Tag, EV empty vector, Mut mutant, OE overexpression, SC scramble, si silencing.

and S2d) and revealed MATα1 Ser114 hyperphosphorylation in human AH livers (Fig. 4d and Supplementary data 1). Altogether, these results demonstrate that MATα1 is hyperphosphorylated at Ser114 in ALD and this could allow increased PIN1-MATα1 interaction to occur.

To examine the importance of Ser114 phosphorylation for PIN1-MATα1 interaction, a MATα1 S144A mutant where Ser114 was substituted with an alanine to prevent its phosphorylation, was generated (Fig. S2e). Histidine (His)-tagged MATα1 WT and S114A were overexpressed in AML-12 and HepG2 cells for 48 h and co-IP analyses using a His-Tag antibody followed by western blot against PIN1 were performed. As shown in Fig. 4e, f, MATα1 phosphorylation at Ser114 is critical for PIN1 to bind to MATα1, as its mutation resulted in a significant loss of their interaction in both cell lines.

**Blocking PIN1-MATα1 interaction protects against alcohol-induced injury by increasing mitochondrial MATα1.** We assessed the effect of Ser114 mutation on MATα1 enzymatic activity by measuring L-methionine consumption and SAMe production using recombinant MATα1 WT and S114A proteins. MATα1 S114A is active and able to produce SAMe but Ser114 mutation reduced MAT activity (Fig. 5a). To evaluate the consequences of blocking PIN1-MATα1 interaction in ALD, MATα1 WT, and S114A were overexpressed in AML-12 cells treated with ethanol. Like endogenous *Mat1a*, ethanol upregulated mRNA levels of MAT1A WT and S114A (Fig. 5b). Interestingly, while ethanol significantly reduced total protein levels of MATα1 WT, it failed to downregulate MATα1 S114A (Fig. 5c, d), supporting the hypothesis that MATα1-S114 is involved in MATα1 down-regulation in ALD. To better understand the effect of ethanol and Ser114 phosphorylation on MATα1 downregulation, a cyclo-heximide (CHX) chase experiment was performed. Ethanol reduced the half-life of MATα1 WT protein by more than half (Fig. 5e), but MATα1 S114A's half-life remained unaltered (Fig. 5e), confirming that the interaction with PIN1 is key for MATα1 protein destabilization by alcohol. Since alcohol increases PIN1-MATα1 interaction and PIN1 inhibits MATα1 mitochondrial targeting, we examined whether blocking the interaction with PIN1 would influence MATα1 mitochondrial targeting in the context of ALD. As shown in Fig. 5f, g, ethanol markedly lowered the mitochondrial MATα1 WT, as seen previously (Fig. 1f, g), but not that of MATα1 S114A. This finding was confirmed by immunofluorescence using TOM70 (Fig. 5h) and ATP5B (Fig. S3a) as mitochondrial markers, which showed co-localization with the Histidine-tagged MATα1 under control condition and ethanol-induced mitochondrial MATα1 depletion in cells expressing WT but not the S114A mutant. We also evaluated MATα1 localization in Golgi apparatus but found no co-localization (Fig. S3b). These findings confirmed that

enhanced PIN1-MATα1 interaction is responsible for the selective depletion of mitochondrial MATα1 content in ALD.

To further analyze how mitochondrial MATα1 contributes to the pathogenesis of ALD, we compared the effect of MATα1 WT versus S114A mutant on mitochondrial function in AML-12 treated with ethanol. By using MitoTracker® Red CMXRos dye which accumulates in active mitochondria, we evaluated the effect of ethanol on mitochondria of MATα1 WT and S114A expressing AML-12 cells. While ethanol reduced the number of active mitochondria in MATα1 WT cells by 50% (Fig. 6a, b), it only reduced it by 25% in MATα1 S114A cells (Fig. 6a, b). In addition, we evaluated membrane potential using Tetramethylrhodamine, Ethyl Ester, Perchlorate (TMRE) dye and found a decrease in MATα1 WT expressing AML-12 cells after ethanol treatment but no changes in cells expressing the S114A mutant (Fig. 6c). We also measured mitochondrial respiration using the Seahorse analyzer and found that the negative effect of ethanol was significantly attenuated by the presence of MATα1 within mitochondria, as observed in S114A expressing AML-12 cells compared to WT (Fig. 6d). The fall in basal respiration, mitochondrial ATP production and maximal respiratory capacity in WT MAT1A expressing cells treated with ethanol were attenuated in cells expressing S114A mutant (Fig. 6d). Protection of mitochondrial function by the S114A mutant was confirmed in AML-12 cells that had endogenous *Mat1a* expression silenced first to eliminate the contribution of endogenous MATα1 (Fig. S3c–g).

Some of the manifestations of ethanol-induced mitochondrial dysfunction include increased production of ROS and a drop in ATP levels[25]. We speculated preserving mitochondrial MATα1 content should protect against these changes and consistently, we found that ethanol lowered ATP levels (Fig. 6e) and increased the production of mitochondrial ROS in MATα1 WT but not in MATα1 S114A expressing cells (Fig. 6f). Last, we assessed the accumulation of triglycerides in AML-12 cells since this is dependent on mitochondrial function and is the first sign of ALD. Ethanol increased the quantity of triglycerides in MATα1 WT cells but had a minimal effect in cells expressing MATα1 S114A (Fig. 6g, h). Altogether, these findings support the importance of mitochondrial MATα1 in preventing ethanol-induced mitochondrial dysfunction and triglyceride accumulation.

**CK2 phosphorylates MATα1 at Ser114 to inhibit its mito-chondrial localization in ALD.** To determine the kinase that phosphorylates MATα1 at Ser114 we used multiple online pre-diction software (Fig. S4a) and evaluated the expression of some of the most highly predicted: p38, GSK3β, JNK, and CK2. We found that CK2 was the only kinase upregulated in all three models of ALD (Fig. 7a–c and Fig. S4b, c). CK2α (the catalytic

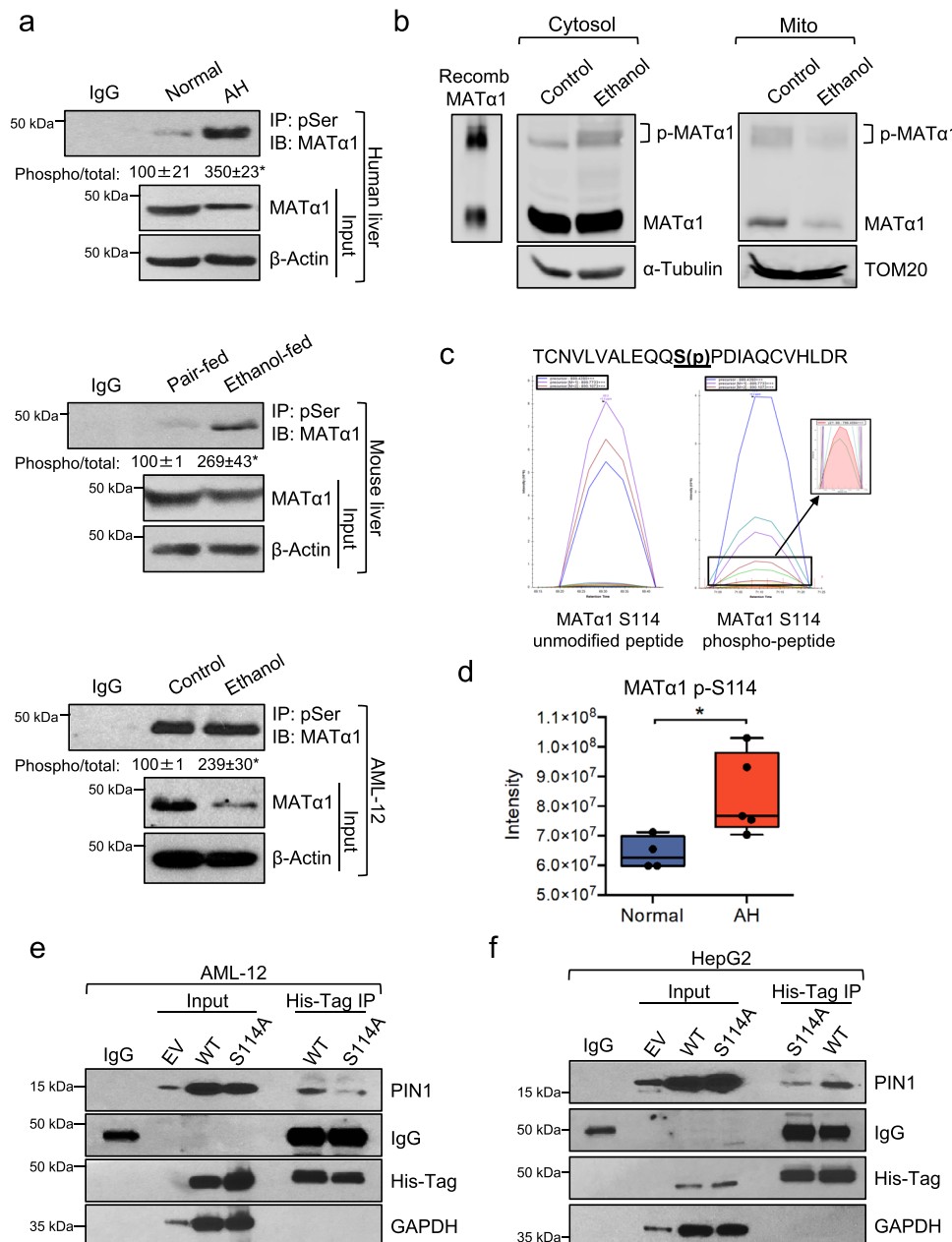

**Fig. 4 Alcohol-induced MATα1 phosphorylation at Ser114 is required for PIN1 binding. a** IP analysis of human normal and AH liver lysates, pair-fed and ethanol-fed mouse livers, and AML-12 cells treated with ethanol using anti-phosphoserine antibody followed by western blot analysis against MATα1 ($n = 3$ independent samples/experiments, $p = 0.001$ for AH vs normal; $p = 0.021$ ethanol-fed vs pair-fed; and $p = 0.008$ ethanol vs control AML-12). **b** Phos-Tag gels of phosphorylated MATα1 in cytosolic and mitochondrial fractions of AML-12 cells treated with ethanol. Unphosphorylated and phosphorylated recombinant human MATα1 protein is shown as a control (left). **c** Area under the curve of the phosphorylated peptide which corresponds to MATα1 Ser114 and its unmodified counterpart in WT mouse liver. **d** Quantitation of the MATα1 Ser114 in normal ($n = 4$) and AH ($n = 5$) human livers ($p = 0.03$). The center line, bounds of box, and whiskers represent mean, 25th to 75th percentile range, and minimum to maximum range. Normal liver is shown in blue and AH in red. **e** IP analysis of AML-12 and **f** HepG2 overexpressing MATα1 WT or S114A cell lysates using anti-His-Tag antibody followed by western blot analysis against PIN1. *$p < 0.05$. Statistical significance was determined by using two-tailed, one-sample *t*-test for treatment comparisons and ANOVA test for group comparisons. Results are shown as mean ± SEM. Source data are provided as a Source data file. AH alcoholic hepatitis, EV empty vector, IP immunoprecipitation, pSer phosphor-serine, WT wild type.

subunit) was found upregulated in human AH (Fig. 7a), the NIAAA mouse model (Fig. 7b), and AML-12 cells treated with ethanol (Fig. 7c), and its activity was doubled by ethanol in the latter (Fig. 7d). We found *CSNK2A1* mRNA levels increased in human AH (Fig. S4d) and NIAAA livers (Fig. S4e), and unchanged in AML-12 cells (Fig. S4f). We confirmed that CK2 phosphorylates MATα1 using recombinant proteins by in vitro kinase assay (Fig. 7e) and found that it does it specifically at

Ser114 as MATα1 S114A could not be phosphorylated (Fig. 7f). *Csnk2a1* knockdown (Fig. S4g) experiments demonstrated that CK2 is involved in ethanol-induced mitochondrial MATα1 depletion, as its knockdown prevented this fall (Fig. 7g, h) and protected against ethanol-induced mitochondrial dysfunction (Fig. 7i–l). Impairment in mitochondrial respiration (Fig. 7i), reduction of ATP (Fig. 7j), increase in mROS (Fig. 7k), and accumulation of triglycerides (Fig. 7l) caused by ethanol were all

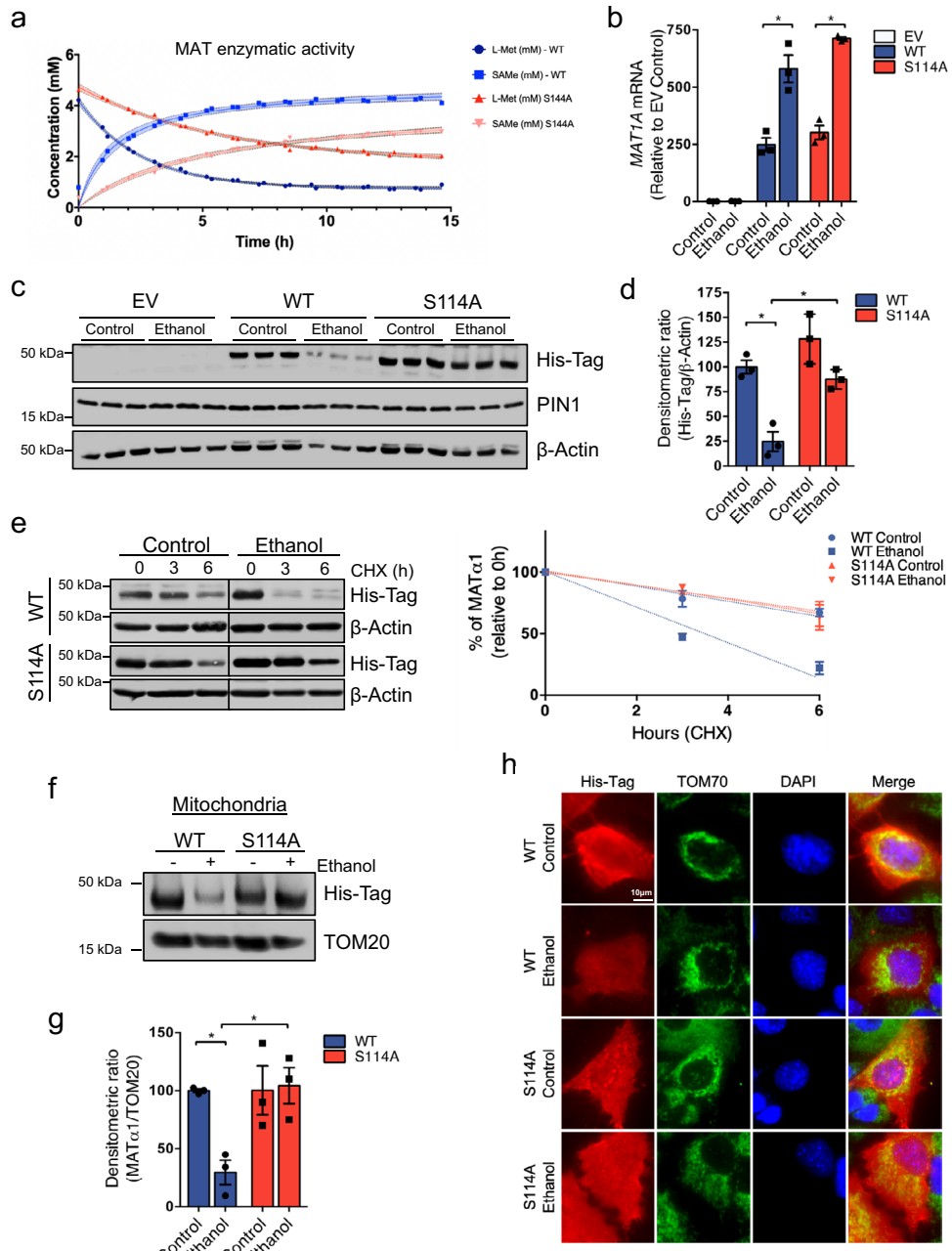

**Fig. 5 Blocking the interaction with PIN1 protects MATα1 against alcohol-induced downregulation. a** MAT enzymatic activity measured as methionine consumption and SAMe production using recombinant MATα1 WT and S114A proteins. **b** *MAT1A* mRNA levels ($n = 3$ independent experiments, $p = 0.007$ for WT ethanol vs control and $p = 0.0002$ S114A ethanol vs control), **c** western blot, and **d** corresponding densitometry analysis of AML-12 cells lysates after transfection with EV, MATα1 WT or S114A vectors and ethanol treatment using anti-His-Tag antibody ($n = 3$ independent experiments, $p = 0.003$ for WT ethanol vs control and $p = 0.005$ S114A ethanol vs WT ethanol). **e** Western blot analysis of AML-12 cells expressing MATα1 WT or S114A and treated with ethanol before CHX addition and graph on the right showing the $t_{1/2}$ for MATα1 WT and S114A with and without ethanol treatment. **f** Western blot and **g** densitometry analysis of the mitochondrial fraction of AML-12 cells expressing MATα1 WT or S114A after ethanol treatment ($n = 3$ independent experiments, $p = 0.003$ for WT ethanol vs control and $p = 0.016$ S114A ethanol vs WT ethanol). **h** His-Tag (red) and TOM70 (green) immunofluorescence and mitochondrial. $*p < 0.05$. Statistical significance was determined by using two-tailed, one-sample $t$-test for treatment comparisons and ANOVA test for group comparisons. Results are shown as mean ± SEM. Source data are provided as a Source data file. WT is shown in blue and S114A is shown in red. CHX cycloheximide, EV empty vector, WT wild type.

attenuated when *Csnk2a1* was silenced and mitochondrial MATα1 content preserved.

**Mitochondrial MATα1 protects mitochondrial proteome against ethanol-induced degradation**. To gain further insight

into the role of MATα1 within mitochondria, we performed MATα1 co-IP + MS analyses in normal human livers. Twenty-one percent of all identified proteins interacting with MATα1 were mitochondrial (Fig. 8a, Supplementary data 2 and 3). To analyze MATα1 mitochondrial interactome, Gene Ontology (GO) and Kyoto Encyclopedia of Genes and Genomes (KEGG)

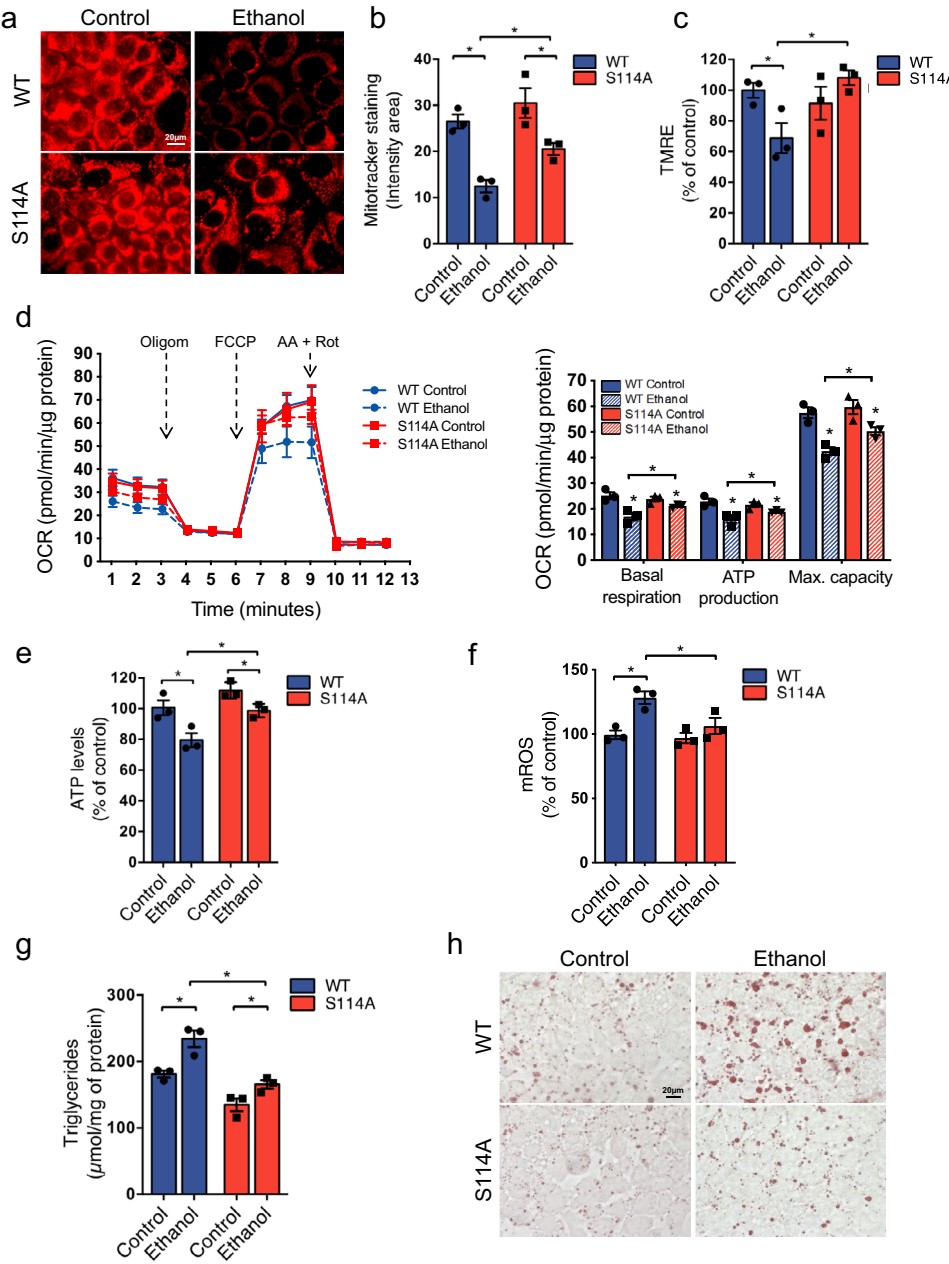

**Fig. 6 Blocking PIN1-MATα1 interaction protects against alcohol-induced mitochondrial injury by increasing MATα1 mitochondrial content. a** Mitotracker staining (red) and **b** quantification ($n = 10$ pictures/3 independent experiments, $p = 0.002$ for WT ethanol vs control; $p = 0.047$ for S114A ethanol vs control; $p = 0.013$ S114A ethanol vs WT ethanol), **c** mitochondrial membrane potential ($n = 3$ independent experiments, $p = 0.045$ for WT ethanol vs control; $p = 0.022$ S114A ethanol vs WT ethanol), **d** mitochondrial respiration and basal respiration ($p = 0.015$ for WT ethanol vs control; $p = 0.049$ S114A ethanol vs control; $p = 0.035$ WT ethanol vs S114A ethanol), ATP production ($p = 0.022$ for WT ethanol vs control; $p = 0.041$ S114A ethanol vs control; $p = 0.041$ WT ethanol vs S114A ethanol) and maximal respiratory capacity ($p = 0.005$ for WT ethanol vs control; $p = 0.040$ S114A ethanol vs control; $p = 0.004$ WT ethanol vs S114A ethanol from 3 independent experiments), **e** ATP ($n = 3$ independent experiments, $p = 0.035$ for WT ethanol vs control; $p = 0.022$ S114A ethanol vs control; $p = 0.041$ WT ethanol vs S114A ethanol), **f** mROS levels ($n = 3$ independent experiments, $p = 0.013$ for WT ethanol vs control; $p = 0.04$ WT ethanol vs S114A ethanol), **g** triglycerides content ($n = 3$ independent experiments, $p = 0.017$ for WT ethanol vs control; $p = 0.008$ WT ethanol vs S114A ethanol), and **h** Oil red O staining in AML-12 cells expressing MATα1 WT or S114A after ethanol treatment. *$p < 0.05$. Statistical significance was determined by using two-tailed, one-sample $t$-test for treatment comparisons and ANOVA test for group comparisons. Results are shown as mean ± SEM. Source data are provided as a Source data file. WT is shown in blue and S114A is shown in red. AA antimycin A, FCCP Carbonyl cyanide-p-trifluoromethoxyphenylhydrazone, mROS mitochondrial reactive oxygen species, OCR oxygen consumption rate, Oligom oligomycin, Rot rotenone, WT wild type.

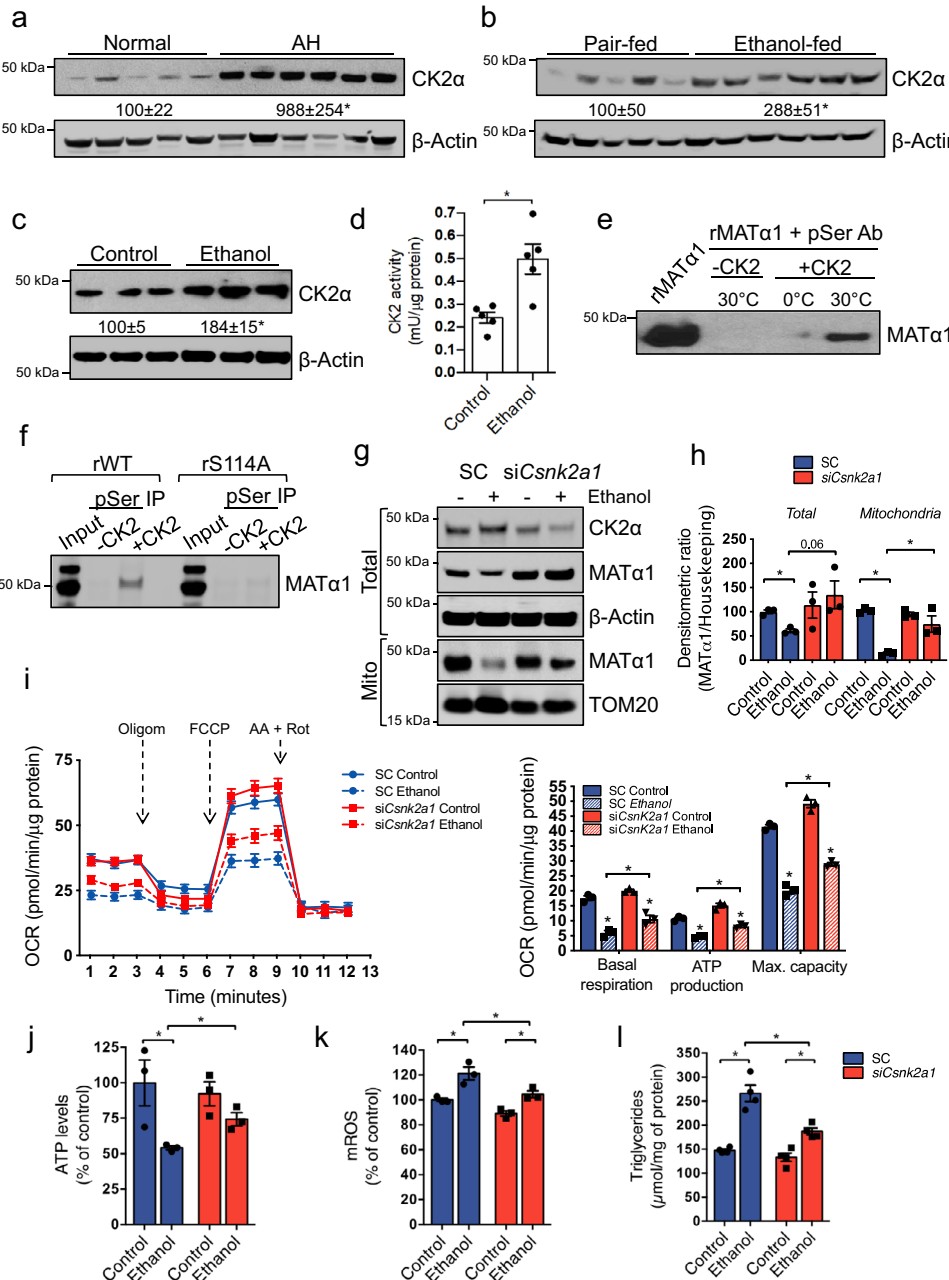

**Fig. 7 CK2 phosphorylates MATα1 at Ser114 to inhibit its mitochondrial localization in ALD.** Western blots and densitometry analyses of CK2α in **a** human normal ($n = 5$) and AH livers ($n = 6$) ($p = 0.017$), **b** pair-fed ($n = 5$) and ethanol-fed ($n = 6$) mouse livers ($p = 0.026$), and **c** AML-12 cells treated with ethanol ($p = 0.005$). **d** CK2 activity in AML-12 cells treated with ethanol ($n = 5$ independent experiments, $p = 0.014$). In vitro kinase assay of CK2 on **e** recombinant human MATα1, and **f** recombinant MATα1 WT and S114A proteins. **g** Western blot and **h** densitometry analyses of total and mitochondrial MATα1 in AML-12 cells after *Csnk2a1* silencing and ethanol treatment ($n = 3$ independent experiments. For mitochondrial MATα1 $p = 0.00002$ SC ethanol vs control; $p = 0.007$ SC ethanol vs si*Csnk2a1* ethanol. For total MATα1 $p = 0.002$ SC ethanol vs control; $p = 0.007$ SC ethanol vs si*Csnk2a1* ethanol). **i** Mitochondrial respiration and basal respiration ($p = 0.004$ for SC ethanol vs control; $p = 0.002$ si*Csnk2a1* ethanol vs control; $p = 0.03$ SC ethanol vs si*Csnk2a1* ethanol), ATP production ($p = 0.0003$ for SC ethanol vs control; $p = 0.001$ si*Csnk2a1* ethanol vs control; $p = 0.002$ SC ethanol vs si*Csnk2a1* ethanol) and maximal respiratory capacity ($p = 0.0009$ SC ethanol vs control; $p = 0.0002$ si*Csnk2a1* ethanol vs control; $p = 0.003$ SC ethanol vs si*Csnk2a1* ethanol), **j** ATP levels ($p = 0.047$ WT ethanol vs control; $p = 0.013$ SC ethanol vs control; $p = 0.041$ SC ethanol vs si*Csnk2a1* ethanol), **k** mROS levels ($p = 0.0017$ WT ethanol vs control; $p = 0.013$ SC ethanol vs control; $p = 0.047$ SC ethanol vs si*Csnk2a1* ethanol), and **l** triglycerides content ($n = 4$ independent experiments, $p = 0.006$ WT ethanol vs control; $p = 0.005$ SC ethanol vs control; $p = 0.041$ SC ethanol vs si*Csnk2a1* ethanol) in AML-12 *Csnk2a1* silenced cells after ethanol treatment. $n = 3$ independent experiments unless specified. *$p < 0.05$. Statistical significance was determined by using two-tailed, one-sample *t*-test for treatment comparisons and ANOVA test for group comparisons. Results are shown as mean ± SEM. Source data are provided as a Source data file. SC is shown in blue and si*Csnk2a1* is shown in red. AA antimycin A, FCCP Carbonyl cyanide-p-trifluoromethoxyphenylhydrazone, IP immunoprecipitation, mROS mitochondrial reactive oxygen species, OCR oxygen consumption rate, Oligom oligomycin, pSer phosphoserine, r recombinant, Rot rotenone, SC scramble, si silencing, WT wild type.

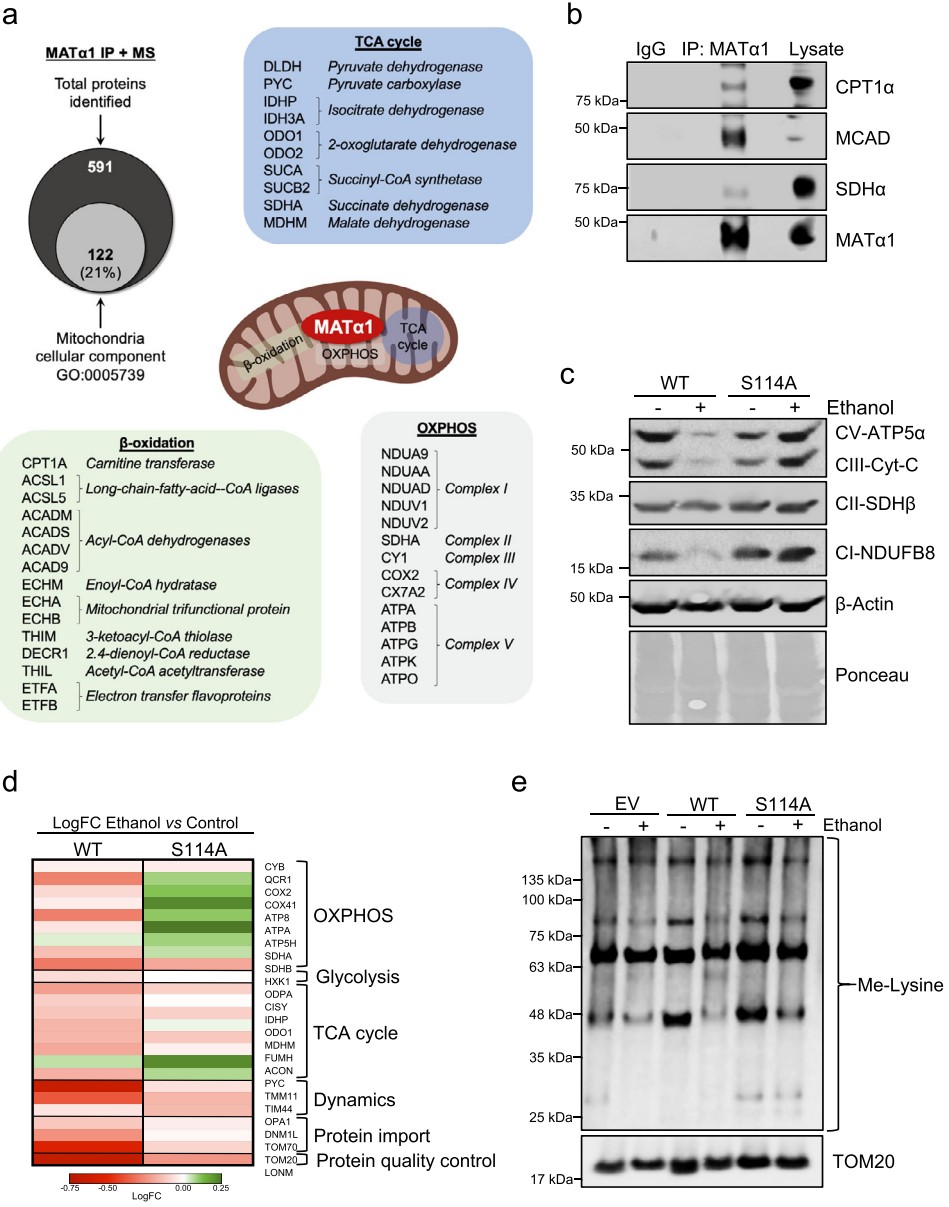

**Fig. 8 Mitochondrial MATα1 protects mitochondrial proteome against ethanol-induced degradation. a** Overlap of mitochondrial and total proteins identified by MS in human livers after MATα1 co-IP and list of mitochondrial proteins involved in TCA cycle, OXPHOS, and fatty acid β-oxidation found to interact with MATα1 in human liver. **b** IP analysis of human normal liver lysates using anti-MATα1 antibody followed by western blot analysis against CPT1α, MCAD, and SDHα. **c** Western blot analysis using an OXPHOS antibody cocktail which detects ATP5A (Complex V), UQCRC2 (Complex III), SDHB (Complex II), and NDUFB8 (complex I) in lysates of AML-12 cells expressing MATα1 WT or S114A after ethanol treatment ($n = 3$ independent experiments). **d** Protein abundance heatmap for MitoPlex proteins detected in AML-12 cells expressing MATα1 WT or S114A after ethanol treatment. Expression is displayed as LogFC of ethanol versus control, ranging from downregulated (red) to upregulated (green) ($n = 4$ independent experiments). **e** Western blot analysis using a methyl-lysine antibody in mitochondrial fractions of AML-12 cells transfected with empty vector (EV), MATα1 WT or S114A after ethanol treatment ($n = 3$ independent experiments). Results are shown as mean ± SEM. Source data are provided as a Source data file. IP immunoprecipitation, LogFC Log2 fold change, MS mass spectrometry, OXPHOS oxidative phosphorylation, TCA tricarboxylic, WT wild type.

pathways enrichment analyses were performed using the STRING database (https://string-db.org). Interestingly, we found enzymes involved in the three main mitochondrial pathways that generate ATP: the TCA cycle, OXPHOS, and fatty acid β-oxidation (Fig. 8a and Fig. S5). We observed that MATα1 interacts with proteins involved in almost every single step of both the TCA cycle and mitochondrial fatty acid β-oxidation, along with all the complexes of the electron transport chain (ETC). The interaction between MATα1 and some of these proteins like CPT1, MCAD, and SDH was validated by regular co-IP in human liver (Fig. 8b). Although not investigated here, proteins involved in other

important pathways including glycolysis, gluconeogenesis, ketone body metabolism, and mitochondrial DNA transcription, were also identified in MATα1's interactome (Supplementary data 2 and 3). These results point to an important role of MATα1 on mitochondrial function and metabolism.

MATα1 co-IP + MS analyses in healthy livers and livers from AH patients showed reduced interaction of MATα1 with 72% of proteins involved in TCA cycle, OXPHOS, and fatty acid β-oxidation (Fig. S5 and Supplementary data 3). The strongest reduction was found between MATα1 and enzymes involved in mitochondrial fatty acid β-oxidation (Fig. S5c), known to be

inhibited in ALD[26]. Bioinformatic analyses using the DAVID software (https://david.ncifcrf.gov) showed that proteins with reduced interaction with MATα1 were mainly mitochondrial and involved in fatty acid β-oxidation, TCA cycle, and OXPHOS. These three pathways, along with glycolysis, gluconeogenesis, synthesis and degradation of ketone bodies, and pyruvate metabolism were all predicted to be altered in AH (Fig. S5d). In addition, the top GO terms—or biological functions—of proteins with reduced interaction with MATα1 were also related to mitochondria and included ATP synthesis, mitochondrial organization, and 2-oxo-glutarate and succinyl-CoA metabolic processes, among many others (Table S1 and Supplementary data 3). Supporting our hypothesis that mitochondrial MATα1 is specifically reduced in ALD, proteins whose interaction with MATα1 was higher in AH were associated with ribosomal, endoplasmic reticulum, proteasomal, and other non-mitochondrial pathways (Table S2 and Supplementary data 3).

Although we do not know whether this reduced interaction is caused by lower mitochondrial MATα1 content or reduced expression of those mitochondrial proteins in ALD, we found that mitochondrial MATα1 positively regulated the expression of several mitochondrial proteins. Previous studies have shown that chronic alcohol consumption alters the expression of mitochondrial proteins[27,28]. OXPHOS respiratory complexes I (NADH dehydrogenase), III (cytochrome b–c1), IV (cytochrome oxidase), and V (the ATP synthase complex) are reduced in multiple models of ALD, impairing mitochondrial respiration[22,29]. Since MATα1 interacts with some subunits of OXPHOS complexes (Fig. 8a) and the interaction with many of them is decreased in human AH (Fig. S5b), we assessed whether mitochondrial MATα1 could influence their expression. By using an anti-OXPHOS cocktail antibody the expression of ETC complexes I to V was evaluated in AML-12 cells expressing either MATα1 WT or S114A mutant after 48 h of ethanol treatment. As shown in Fig. 8c, ethanol reduced the expression of complexes I, III, and V in MATα1 WT expressing cells. However, MATα1 S114A cells, where mitochondrial MATα1 is preserved, were resistant to the drop in OXPHOS complexes (Fig. 8c). Further analysis using MitoPlex, a high throughput quantitative MS-based assay of proteins critical to central carbon metabolism and overall mitochondrial function[30] confirmed this finding (Supplementary data 4). While OXPHOS proteins were downregulated in MATα1 WT AML-12 cells, they were not only resistant to that drop, but increased in MATα1 S114A AML-12 cells treated with ethanol (Fig. 8d). The downregulation of proteins involved in the TCA cycle, mitochondrial dynamics, and protein import and quality control was also attenuated in MATα1 S114A expressing cells (Fig. 8D). Supporting our hypothesis that MATα1 provides SAMe within mitochondria for methylation, we found increased lysine-methylation of mitochondrial proteins in MATα1 S114A expressing cells where MATα1 mitochondrial content and expression of mitochondrial proteins are preserved (Fig. 8e). Importantly, at least 69% of the mitochondrial proteins that interact with MATα1 have known methylation sites (Supplementary data 2). Altogether these results confirmed that mitochondrial MATα1 is critical in maintaining many essential mitochondrial proteins and protects against ethanol-induced mitochondrial dysfunction.

## Discussion

ALD is a major health burden. Globally, ~2 billion people consume alcohol and 75 million have alcohol-use disorders, including ALD[2]. ALD is one the most prevalent and devastating conditions caused by alcohol and one of the leading causes of alcohol-related deaths[2]. Although there have been numerous advances in the understanding of alcohol-related liver diseases, specific treatment for ALD remains elusive. Over the years, clinical trials have attempted to ameliorate different factors of the pathogenesis of ALD, like inflammation, oxidative stress, and nutritional abnormalities, but overall, little to no improvement was found in those studies[31]. There are no FDA-approved therapies for ALD; lifestyle changes including abstinence and nutritional support are the only options that can improve outcomes.

SAMe is the main methyl donor and a major regulator of key biological pathways in the cell. Besides being used for methylation of RNA, DNA, and proteins, SAMe is a precursor of GSH, playing a major antioxidant role. It is essential for life and found in all organs, fluids, and cellular organelles. The liver is considered the body's SAMe factory and SAMe levels are tightly regulated for normal liver function[4]. SAMe is reduced in patients with AH[32] and murine models of ALD[33,34] and its administration showed promising effects against the progression of the disease in both humans[35] and mice[29,36]. Two years of SAMe administration to patients with less advanced alcoholic cirrhosis resulted in an overall decline in mortality compared to placebo[35]. The human trial only showed significance in a post hoc analysis so whether SAMe is an effective agent in ALD remains to be confirmed. In ethanol-fed rats, SAMe has been shown to protect mitochondria and reduce liver injury[29,36].

SAMe is present in mitochondria and until recently, the mitochondrial SAMe transporter, encoded by *SLC25A26*, was believed to be the only route of SAMe entry into the organelle[37]. Our previous study showed that MATα1, enzyme responsible for SAMe synthesis in the liver, also localizes to mitochondria of hepatocytes[3]. Although little is known, methylation within the mitochondria is attracting increasing attention. It was recently discovered that human mtDNA is extensively methylated[38] and suggested that methylation plays a role in mtDNA gene expression and replication[39,40]. Several mitochondrial protein methyltransferases have been identified[41–43] and an increasing number of mitochondrial proteins are found to be methylated[41–43]. All these findings suggest methylation is an important modification within mitochondria. A recent publication[44] shows that mitochondrial SAMe deletion negatively impacts OXPHOS and TCA cycle metabolism and leads to a profound mitochondrial defect affecting both mitochondrial translation and OXPHOS with both nuclear and mitochondrial encoded subunits of complexes I, III, and IV severely decreased. This study concluded that SAMe transporter is the only source of SAMe in mitochondria but none of the studies were performed in cells that express MAT1A. Our results demonstrate that MATα1 is another source of mitochondrial SAMe in the liver and that mitochondrial MATα1 is critical to preserve mitochondrial proteins and function likely via local SAMe production. Since SAMe has a short half-life and is needed to methylate numerous substrates, having MATα1 in close proximity ensures a source of SAMe for key substrates that are regulated by methylation. Consistently, majority of the mitochondrial proteins that interact with MATα1 are known to be methylated.

Like other highly metabolic active organs, the liver is very rich in mitochondria and is dependent on their function[45]. Alcohol alters the structure and function of hepatic mitochondria[7] and due to their essential role in ATP production, metabolism, and cell death, these changes are important contributory factors in ALD[7]. This prompted us to investigate mitochondrial MATα1 in ALD and led us to uncover a mitochondrial mechanism that contributes to ALD pathogenesis that could provide new targets for treatment.

Our study used two experimental models and two human ALD stages. The NIAAA model consists of a chronic-plus-single-binge

feeding and induces liver injury, inflammation, and fatty liver, mimicking acute-on-chronic alcoholic liver injury in patients[22]. The model of AML-12 cells treated with ethanol recapitulates only the early stage of ALD but allows molecular dissection of the effect of ethanol on hepatocytes and its organelles. Key changes from these models were confirmed using human AH and end-stage cirrhotic ALD samples, which are limited in quantity. Using this approach, we found mitochondrial MATα1 is selectively reduced in ALD, as seen in human AH liver samples, NIAAA model, and AML-12 cells treated with ethanol. More importantly, preserving mitochondrial MATα1 protected against alcohol-induced mitochondrial injury and triglyceride accumulation, suggesting that mitochondrial MATα1 depletion is important in ALD pathogenesis. While investigating the molecular mechanism of MATα1 mitochondrial depletion we focused on PIN1 because MATα1 has a PIN1 motif and PIN1 has been shown to alter the mitochondrial translocation of some targets, thereby regulating mitochondrial function[9], oxidative stress[9,10], and apoptosis[11,12]. Indeed, we found that MATα1 is a target of PIN1, as both proteins interact in liver, and further analyses in human AH livers, livers of ethanol-fed mice, and ethanol-treated AML-12 cells showed that PIN1-MATα1 interaction is enhanced in ALD. Although PIN1 did not alter MATα1 expression, it regulated its mitochondrial localization. *PIN1* silencing in AML-12 and HepG2 cells raised mitochondrial MATα1 content, and its overexpression reduced it, demonstrating that PIN1 negatively regulates MATα1 mitochondrial content via isomerization, as the catalytic mutant of PIN1 had no such effect. Along with increased mitochondrial MATα1 content, *Pin1* silencing increased ATP levels and reduced triglycerides in AML-12 cells after ethanol treatment. PIN1 has many targets and its silencing led to enhanced fatty acid β-oxidation and attenuated hepatic fat accumulation in a mouse model of NASH[15], an improvement in mitochondrial function may only be partly due to MATα1.

To study the importance of MATα1-PIN1 interaction in ALD, we inhibited it by generating a MATα1 mutant lacking Ser114, the only PIN1 binding motif. MATα1 Ser114 was confirmed to be hyperphosphorylated in human AH livers compared to controls, which correlates with enhanced PIN1-MATα1 interaction in ALD. Ser114 mutation to alanine -to prevent its phosphorylation- inhibited PIN1-MATα1 interaction in both AML-12 and HepG2 cells, confirming that Ser114 is key for PIN1 to bind to MATα1. It is noteworthy that these experiments do not allow us to determine the stoichiometry of Ser114, which may also play a role in MATα1 interaction with PIN1 and mitochondrial targeting.

MATα1 S114A allowed us to study the effect of PIN1 on MATα1 specifically. This is important since PIN1 has many other targets. We found PIN1 is directly involved in ethanol-induced MATα1 protein downregulation, as MATα1 S114A mutant was protected from degradation. Interestingly, PIN1 binding to MATα1 was also associated with the reduction in MATα1's half-life, as MATα1 S114A's remained unchanged after ethanol treatment. Even though how PIN1-MATα1 interaction facilitate MATα1 degradation remains to be determined, its mitochondrial targeting prevented it from degradation. More importantly, preserving MATα1 mitochondrial content protected against alcohol-induced mitochondrial dysfunction. Higher mitochondrial MATα1 content attenuated alcohol-induced mROS generation, reduction in ATP levels, and improved overall mitochondrial function, which translated to lower triglyceride accumulation. We identified CK2 as a kinase that phosphorylates MATα1 at Ser114. Although there might be other kinases that phosphorylate this residue, we found that CK2 expression and activity are higher in ALD, and its silencing prevented ethanol-induced mitochondrial MATα1 depletion and protected from mitochondrial injury,

suggesting a role regulating MATα1 mitochondrial localization in ALD.

The next question is how mitochondrial MATα1 protects against ethanol-induced mitochondrial dysfunction. Our main hypothesis is that mitochondrial MATα1 provides an additional source of SAMe within the organelle to regulate mitochondrial methylation and function. Supporting this, we found that in normal human liver, MATα1 interacts with proteins that participate in important mitochondrial metabolic pathways—the TCA cycle, OXPHOS, and fatty acid β-oxidation. As shown in Fig. 8, we identified proteins involved in almost every step of all three pathways, suggesting a global role of MATα1 in regulating them. Proteins involved in other pathways such as pyruvate metabolism, ketone bodies, and amino acid metabolism were also identified, which suggests an even broader mitochondrial function regulation by MATα1. By further MATα1 IP + MS analyses in normal and AH human livers, we compared MATα1 mitochondrial interactome and found a selective reduction in the interaction of MATα1 with mitochondrial proteins in AH livers. This reduction is selective, as there is increased interaction of MATα1 with many proteins, but none of them are mitochondrial or associated with any mitochondrial pathway.

Alcohol has a negative impact on many mitochondrial proteins[27,28]. Besides inhibiting mitochondrial protein synthesis machinery[46], alcohol may also induce the irreversible oxidation of mitochondrial proteins[47]. To investigate whether mitochondrial MATα1 could regulate the expression of mitochondrial proteins, we evaluated components of the ETC. Along with mitochondrial MATα1, ethanol reduced the protein levels of subunits of complexes I, III, and V. However, when mitochondrial MATα1 was preserved, these proteins were preserved as well. Using MitoPlex, a MS-based technique that assesses the expression of mitochondrial proteins, we confirmed that mitochondrial MATα1 not only prevented the reduction of many mitochondrial proteins involved in TCA cycle and OXPHOS, but also increased the expression of some of them. Mitochondrial MATα1 also protected proteins involved in glycolysis, mitochondrial dynamics, protein import, and protein quality control from ethanol-induced fall. Whether MATα1 prevents ethanol-induced mitochondrial protein degradation or regulates mitochondrial proteins translation or biogenesis remains to be studied. Correlating with higher mitochondrial MATα1, we found increased mitochondrial protein methylation in S114A expressing hepatocytes, strongly supporting our hypothesis that MATα1 localization within mitochondria helps maintain its methylation status and function.

Our study is limited by lack of mitochondrial SAMe levels to substantiate increased SAMe availability within the mitochondria in S114A mutant. We attempted repeatedly to measure mitochondrial SAMe and GSH levels but the results were inconsistent from one experiment to another. This raises the possibility that metabolites may leak out from damaged mitochondria, caused by alcohol in this case. Another limitation is that we have not proven mice expressing MATα1 S114A mutant is protected from ALD. This is planned for future experiments.

In summary, we show that mitochondrial MATα1 is important for hepatic mitochondrial function and that its depletion occurs early and persists to end-stage ALD through a mechanism that involves its phosphorylation at Ser114 by CK2 and interaction with PIN1 (Fig. 9). Blocking the interaction between PIN1 and MATα1 and silencing *Csnk2a1* raised mitochondrial MATα1 and attenuated alcohol-induced mitochondrial injury and triglyceride accumulation. This data supports a targetable mechanism by which alcohol induces mitochondrial injury. These findings also illustrate the importance of mitochondrial MATα1 in mitochondrial metabolic processes.

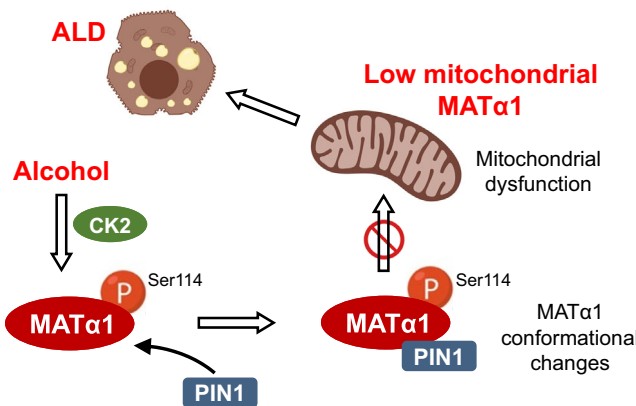

**Fig. 9 PIN1 impairs MATα1 mitochondrial targeting iEn alcohol-associated liver disease.** Graphical summary of the study. Alcohol promotes CK2 phosphorylation of MATα1 at Ser114 and interaction with PIN1, thereby inhibiting MATα1 mitochondrial targeting. Alcohol-induced mitochondrial MATα1 depletion contributes to mitochondrial dysfunction and the pathogenesis of ALD.

## Methods

**Human samples.** Liver specimens from subjects with ALD (n = 5) were obtained from the explant during liver transplantation. All had history of alcohol consumption averaging at least 80 g/day for men or 50 g/day for women, for at least 10 years. This criterion is based on epidemiological evidence of the alcohol consumption and cirrhosis relationship. We excluded patients with hepatitis B or C, autoimmune liver disease, hemochromatosis, Wilson disease, and HCC. In addition, liver specimens from subjects with severe alcoholic hepatitis (AH) (n = 6) were obtained from Johns Hopkins University from the explant during liver transplantation. Liver samples from controls were from subjects with no known history of excessive alcohol use or underlying chronic liver disease. They were obtained during abdominal surgeries for various causes. All the samples were received in frozen state and stored at −80 °C until analysis. All liver samples were obtained under a protocol approved by the Human Subjects and Institutional Review Board at Indiana University Indianapolis and Johns Hopkins University. Written informed consent was obtained from each participant.

**Mice.** Four-month-old female C57BL/6 WT littermates were used for this study. Mice were treated with ethanol or pair-fed control diet following the protocol of the National Institute on Alcohol Abuse and Alcoholism (NIAAA) model[22] (protocol 7952). Briefly, mice were fed ad libitum with the Lieber-DeCarli liquid diet (Bio-Serv) for 5 days and then divided into two groups: the ethanol group (n = 6) was fed a liquid diet containing 5% ethanol for 10 days and the control group (n = 5) was pair-fed control diet for 10 days. At day 11, mice in the ethanol group received a single-binge ethanol feeding (5 g/kg, 20% ethanol) while mice in the control group were received dextrin maltose. The gavage was performed early in the morning and after gavage, mice were kept on control or ethanol diet and in the cages with water. The mice were euthanized 9 h after the gavage. All procedure protocols, use, and the care of the animals were reviewed and approved by the Institutional Animal Care and Use Committee at Cedars-Sinai Medical Center and all experiments involving research animals were conducted in accordance with all relevant ethical regulations. All mice are housed under 12-h light/12-h dark cycle at an average temperature of 74 F and 40% humidity.

**Cells.** AML-12 cells were grown in DMEM-F12 medium (Corning) containing a cocktail of insulin, selenium, and transferrin (41400-045 Thermo Scientific), 0.05 μg/mL dexamethasone (D4902 Sigma), 10% serum and antibiotics. Cells were treated with ethanol (E7023 Sigma) 100 mM (twice a day) for 48 h. AML-12 cells were treated with MG132 10 μM and chloroquine (CQ) 50 μM 6 h and 16 h before harvesting, respectively. HepG2 cells were grown in DMEM medium (Corning) containing 10% serum and antibiotics. After the incubation period, RNA or protein was isolated for various assays described below. AML-12 (Cat.: CRL-2254) and HepG2 (HB-8065) cells were purchased from ATCC. We used HepG2 cells in two figures, to show PIN1 expression alters the mitochondrial content of MATα1 in a human liver cell line in Fig. 3 and that S114A MAT1A mutant interacts much less with PIN1 in Fig. 4. This cell line was authenticated in 2014 by STR testing and no anomalies were detected. We have used this cell line extensively because of ease of transfection and more importantly because it expresses MAT1A.

**Cell lysis and western blotting.** For western blotting, 10–30 μg of whole-cell extract, cytoplasmic protein, or mitochondrial extract were separated by 10% SDS-PAGE. Blots were probed with antibodies diluted 1:1000 in 5% skim milk against

MATα1 (ab129176), PIN1 (07-091, Millipore), TOM20 (ab78547), DDK (TA50011-100, Origene), His-Tag (A00186-100, GenScript), OXPHOS (ab110413), CK2α (2656, Cell Signaling), CPT1α (12252, Cell Signaling), SDHα (5839, Cell Signaling), MCAD (ab110296), JNK (9252, Cell Signaling), pJNK (4668, Cell Signaling), p38 (9212, Cell Signaling), p-p38 (9211, Cell Signaling), GSK3β (9315, Cell Signaling), p-GSK3β (9336, Cell Signaling), methyl-lysine (ab23366), α-Tubulin (32-250022125, Invitrogen), β-actin (A3854 Sigma), and GAPDH (5174 Cell signaling Technology). HRP-linked secondary antibodies anti-rabbit (7074S Cell Signaling) and anti-mouse (7076S Cell Signaling) were diluted 1:5000 in 5% skim milk.

**Phos-tag.** 40 μg of cytosolic and 10 μg of mitochondrial protein from AML-12 cells were prepared using EDTA-free radioimmunoprecipitation assay buffer (RIPA) and separated on 40 μM-phostag gels. PhosTag™ molecule (Fujifilm Wako Chemicals, Richmond, VA) was bound to zinc ions in a neutral gel system. Separated proteins were immunoblotted with MATα1 antibody.

**Subcellular fractionation.** To extract mitochondria from liver the Mitochondria Isolation Kit for Tissue (ab110168) was used following manufacturer's instructions. Briefly, 50–100 mg of livers were washed twice with 1.5 mL of Wash Buffer. Tissue was minced, placed in pre-chilled Dounce homogenizer and homogenized (30 strokes) in 1 mL of isolation buffer. Homogenates were taken to 2 mL with isolation buffer and centrifuged at 1000 × g for 10 min at 4 °C. Supernatants were centrifuged at 12,000 × g for 15 min at 4 °C. Supernatant (cytosolic fraction) was collected and pellets (mitochondrial fraction) were washed with 1 mL of isolation buffer and centrifuged one more time at 12,000 × g for 15 min at 4 °C. Pellets were collected and resuspended in isolation buffer supplemented with protease inhibitor cocktail. For AML-12 and HepG2 cells, mitochondria were extracted using the Mitochondria Isolation Kit for cultured cells (Thermo Scientific) following the protocol provided.

**Immunoprecipitation (IP) analysis.** 700 μg of whole liver, cells extracts, or cytosolic fractions were pre-cleaned by adding 0.7 μg of the appropriate normal Ig together with 20 μl of appropriate protein A + G-agarose conjugate (Santa Cruz Biotechnology, Dallas, TX) for 1 h at 4 °C. The IPs were performed with 3 μg of anti-PIN1 or anti-MATα1 antibody overnight at 4 °C. However, for the mass spectrometry (MS) studies, a total of 1000 μg whole-cell lysate was used to co-IP with antibody against MATα1. In all cases, the protein complexes bound to the protein A + G-agarose conjugate were washed five times with IP buffer (150 mM NaCl, 0.5% deoxycholate, 0.1% SDS, 1% NP-40, 50 mM Tris-HCl, pH 7.5) with protease inhibitors and then diluted in IP buffer (50 mM Tris-HCl, 150 mM NaCl, 2 mM EDTA, 2 mM EGTA, 25 mM NaFl, 25 mM β-glycerophosphate (pH 7.5), 0.1 mM sodium orthovanadate, 0.1 mM PMSF, 5 μg of leupeptin per ml, 0.2% (vol/vol) Triton X-100, 0.5% (vol/vol) Nonidet P-40). Sample was processed for MS as outlined below or separated by 10% SDS-PAGE for immunoblot. The gel was transferred to nitrocellulose membrane and probed with antibodies as described above. Clean Blot IP Detector Reagent (21230, Thermo Fisher) was used to reduce background. Normal Ig (Cell Signaling) was used as a control.

**Direct protein–protein interaction.** Recombinant human MATα1 and PIN1 proteins were from ProSpec. Two micrograms of MATα1 or PIN1 recombinant protein was immobilized to agarose beads by their respective antibody. After washing, beads were mixed with 1 μg MATα1 or PIN1 protein and rotated for 4 h at 4 °C. Beads were then washed six times in binding buffer (50 mM Tris-HCl pH 7.5, 0.2 mm EDTA, 0.25 mM PMSF, 0.5% NP-40), boiled in SDS sample buffer, and proteins separated on 10% SDS-PAGE subjected to western blotting. IgG was used as negative control.

**On-bead digestion and proteomic preparation and protein identification using mass spectrometry.** The co-IP sample underwent on-bead trypsin/Lys-C digestion[3]. Briefly, the protein complex bound to the anti-MAT α1 antibody-protein A + G-agarose IP were denatured and reduced by incubation in urea (2 M) in Tris-HCl (50 mM, pH 7.5) solution containing DTT (15 mM) for 1 h at room temperature. The samples were alkylated with the addition of iodoacetamide (30 mM) for 30 min at room temperature in the dark and incubated with 1 μg of Trypsin/Lys-C mix (Promega) for overnight on a shaker at 37 °C. Digestion was quenched with 1% TFA. The samples were briefly centrifuged and the supernatant was desalted using HLB plates (Oasis HLB 30 μm, 5 mg sorbent, Waters).

**Mass spectrometry acquisition methods.** (DDA-MS) Data-dependent acquisitions and data-independent acquisition mass spectrometry (DIA-MS) acquisitions were performed on an Orbitrap LUMOS Fusion mass spectrometer equipped with an EasySpray ion source and connected to an Ultimate 3000 nano LC system with a 60-min gradient[48]. Peptides were loaded onto a PepMap RSLC C18 column (2 μm, 100 Å, 150 μm i.d. × 15 cm, Thermo) using a flow rate of 1.4 μL/min for 7 min at 1% B (mobile phase A was 0.1% formic acid in water and mobile phase B was 0.1% formic acid in acetonitrile) after which point, they were separated with a linear gradient of 5–20%B for 45 min, 20–35%B for 15 min, 35–85%B for 3 min, holding

at 85%B for 5 min, and re-equilibrating at 1%B for 5 min. Each sample was followed by a blank injection to both clean the column and re-equilibrate at 1%B. The nano-source capillary temperature was set to 300 °C and the spray voltage was set to 1.8 kV. For DIA analysis, MS1 scans were acquired in the Orbitrap at a resolution of 60,000 Hz from mass range 400–1000 $m/z$. For MS1 scans the AGC target was set to $3 \times 10^5$ ions with a max fill time of 50 ms. DIA MS2 scans were acquired in the Orbitrap at a resolution of 15,000 Hz with fragmentation in the HCD cell at a normalized CE of 30. The MS2 AGC was set to 5e4 target ions and a max fill time of 22 ms. DIA was performed using 4 Da (150 scan events) windows over the precursor mass range of 400–1000 $m/z$ and the MS2 mass range was set from 100 to 1500 $m/z$. For DDA analysis MS1 scans the AGC target was set to $4 \times 10^5$ ions with a max fill time of 50 ms. MS2 spectra were acquired using the TopSpeed method with a total cycle time of 3 s and an AGC target of $5 \times 10^4$ and a max fill time of 22 ms, and an isolation width of 1.6 Da in the quadrupole. Precursor ions were fragmented using HCD with a normalized collision energy of 30% and analyzed in the Orbitrap at 15,000 K resolution. Monoisotopic precursor selection was enabled and only MS1 signals exceeding 50,000 counts triggered the MS2 scans, with +1 and unassigned charge states not being selected for MS2 analysis. Dynamic exclusion was enabled with a repeat count of 1 and exclusion duration of 15 s. These methodologies were consistent with those used in Robinson et al.[48] and were identical for the phosphorylation site mapping experiment and the MATα1 interactome analysis.

**Mass spectrometry data analysis.** For binding partner data analysis, DIA-MS results were assayed against the murine NASH library we described[48]. Peak group extraction and FDR analysis were done as we outlined[49]. Briefly, raw intensity data for peptide fragments was extracted from DIA files using the open-source SWATH workflow automated targeted analysis of data-independent acquisition MS data against the sample-specific peptide assay. Then, retention time prediction was made using the Biognosys iRT Standards spiked into each sample. Target and decoy peptides were then extracted, scored, and analyzed using the mProphet algorithm (PyProphet version 2.1.2) to determine scoring cutoffs consistent with 1% FDR. Peak group extraction data from each DIA file was combined using the 'feature alignment' script, which performs data alignment and modeling analysis across an experimental dataset. Finally, all duplicate peptides were removed from the dataset to ensure that peptide sequences are proteotypic to a given protein in our FASTA database.

For Ser114 phosphorylation analysis, first, spectra were obtained from DDA-MS acquisitions of a MATα1 immunoprecipitation. Then, Raw DIA-MS data was imported into Skyline (Skyline Daily, version 19.1.1.309)[50], which was then used to visualize and manually validate the precursor area under the curve (XIC) of the Ser114 phosphorylated and unmodified peptides. A co-eluting precursor and fragment trace was observed consistent with phosphorylation of MATα1 Ser114.

**Site-directed mutagenesis and DNA sequencing.** For the generation of MATα1 S114A mutant, primers were purchased from Eurofins Genomics. Primers were as follows: S114A Rev: 5′-GCAATATCTGGGGCTTGCTGCTCCAAAG-3′ and S114A Fwd: 5′-CTTTGGAGCAGCAAGCCCCAGATATTGC-3′. For the generation of PIN1 R68A/R69A mutant, primers were purchased from Eurofins Genomics. Primers were as follows: PIN1 R68A/69A Rev: 5′-GGTGAAGCACAGCC AGTCAGCGGCGCCCCTCGTCC-3′ and PIN1 R68A/69A Fwd: 5′-GGACGAG GGCGCCGCTGACTGGCTGTGCTTCACC-3′. The quick-change lightning multi-site-directed mutagenesis kit (Agilent Technologies) was used to generate mutants. Mutations were confirmed through sequencing by GENEWIZ.

**Gene overexpression.** For gene overexpression experiments, $7.5 \times 10^4$ (6-well plate) or $2 \times 10^6$ (10 cm dish) HepG2 cells and $1 \times 10^5$ (6-well plate) or $1.5 \times 10^6$ (10 cm dish) AML-12 cells were transiently transfected with His-Tagged-MAT1A (GeneCopoeia), His-Tagged-MAT1A S114A mutant, Myc-DDK-tagged PIN1 (Origene), Myc-DDK-tagged PIN1 R68A/69A overexpression vectors or empty vector using JetPrime® (Polyplus) according to the manufacturer's protocol. 1.6 μg of target plasmid per 6-well plate and 8 μg per 10 cm dish were used for transfection overnight. Next morning medium was changed to normal medium. The cells were cultured for additional 48 h for protein expression mRNA and analysis.

**Gene silencing.** For PIN1 knockdown 50 nM siRNA against human PIN1 (SR303529, Origene) or mouse Pin1 (SR403748, Origene) or equivalent scramble control (SC) were delivered into HepG2 or AML-12 cells by Lipofectamine RNAiMAX (Life Technologies) for 48 h following the manufacturer's protocol. For Csnk2a1 and Mat1a knockdown, 50 nM siRNA against mouse Csnk2a1 (s64538, Thermo Scientific), mouse Mat1a (AM16708, Thermo Scientific), or equivalent scramble control (SC) were delivered into AML-12 cells following the same protocol described above. The same number of cells was used as for gene overexpression.

**MAT protein purification and activity measurement.** MAT1A WT and MAT1A S114A mutant construct (pET-15b) were provided by GenScript. Both were overexpressed in LB medium in Escherichia coli BL21. Protein expression was induced with 0.5 Mm isopropyl-b-D-thiogalactopyranoside and incubated overnight at 20 °C. The

cell pellets were resuspended in lysis buffer (50 mM Tris, 150 mM KCl, 20 mM MgCl₂, 0.1 mM DTT pH 8). Cell disruption was by sonication (60% Amplitude 7:30 ON 7:30 OFF) and cell suspension was centrifuged at $137,000 \times g$ for 30 min at 4 °C. The supernatant was loaded onto a Ni-NTA resin column (Invitrogen) pre-equilibrated with wash buffer (50 mM Tris, 150 mM KCl, 20 mM MgCl₂, 0.1 mM DTT, 20 mM Imidazole pH 8). The proteins were eluted with elution buffer (50 mM Tris, 150 mM KCl, 20 mM MgCl₂, 0.1 mM DTT, 500 mM Imidazole pH 8). Finally, the proteins were loaded into a Hi-Load 16/600 Superdex 200 gel-filtration column, GE Healthcare pre-equilibrated with 50 mM Tris, 150 mM KCl, 20 mM MgCl₂, 1 mM DTT pH 8. Protein quantification was performed by spectrophotometrical measurement (EMAT1A-280 nm) = 40,000 $M^{-1}$ cm$^{-1}$).

MAT enzymatic activity was measured in NMR spectroscopy on a Bruker 600 MHz (12 T) Avance III spectrometer equipped with a BBO probehead at 310 K. NMR sample was composed of 2.4 μM MATα1 WT or MATα1 mutant, 10 mM ATP, 5 mM L-Methionine-(methyl-13C) in the reaction buffer (50 mM Tris, 150 mM KCl, 20 mM MgCl₂, 1 mM DTT pH 8). 1 mM Sodium trimethylsilylpropanesulfonate (DSS) was used as internal reference. A 1D-1H Carr-Purcell_Meiboom_Gill (CPMG) pulse sequence with water signal suppression (CPMGPR1D) (6 min) and a 1D-13C zgpg30 (10 min) spectra were collected for >12 h in order to monitor the substrate disappearance (ATP and L-Met-(methyl-13C)) and product synthesis (SAMe-(methyl-13C)) of the MATα1 enzyme. Chemical shifts were assigned by spike. Data analysis was performed using MestReNova. All transformed spectra were corrected for phase and baseline distortions and referenced to DSS singlet at 0 ppm. Quantification was performed by referencing peak integrals to DSS.

**Protein stability assay and half-life determination.** For CHX chase experiments, AML-12 cells were transfected with His-Tag-MATα1 WT or His-Tag-MATα1 S114A and treated with ethanol for 48 h. The media was changed to serum free prior to the addition of 10 μg/mL CHX for 0, 3, and 6 h. Protein levels were determined at the indicated time points by western blotting as described previously using anti-His-Tag antibody. The relative amount of MATα1 protein was evaluated by densitometry and normalized to β-actin. Protein half-life was determined for each experiment using linear regression analysis.

**Gene expression analysis.** Total RNA was isolated with Trizol (Invitrogen). Total RNA (1–2 μg) was reverse transcribed into cDNA using M-MLV Reverse Transcriptase (Invitrogen). Two microliters of RT product were subjected to quantitative real-time PCR analysis. TaqMan probes for human MAT1A (Hs01547962_m1), PIN1 (Hs01598308), and CSNK2A1, and murine Mat1a (Mm00522563_m1), Pin1 (Mm03053328_s1), and Csnk2a1 (Mm00786779_s1) were purchased from Applied Biosystems. Universal PCR Master Mix was purchased from Bio-Rad. Ubiquitin C was used as housekeeping gene (Hs01871556_s1 and Mm02525934_g1). The cycle Ct value of the target genes was normalized to that of the housekeeping gene to obtain the delta Ct (ΔCt). The ΔCt obtained was used to find the relative expression of target genes according to the formula: relative expression = 2 − ΔΔCt, where ΔΔCt = ΔCt of target genes in experimental groups −ΔCt of target genes in control group.

**Mitochondrial respiration.** Intact cellular respirometry was conducted using a Seahorse XF e96 extracellular flux analyzer. AML-12 cells were seeded (10,000 cells/ well) onto 96-well XF cell culture plates, and transfected using the protocol described above. Cells were treated with ethanol for 72 h and the day of the assay they were refreshed with bicarbonate-free DMEM containing 5.5 mM glucose, 1 mM sodium pyruvate, 4 mM glutamine, and equilibrated for 1 h at 37 °C in a non-CO₂ incubator. Oxygen consumption was subsequently monitored following sequential injection of oligomycin (4 μM), FCCP (1 μM) & antimycin/rotenone (2 μM/2 μM). For normalizing respiration rates, cells were subsequently lysed in lysis buffer and protein concentration was determined using the BCA assay.

**Mitotracker.** Active mitochondria were measured using MitoTracker® Red CMXRos (9082 Cell Signaling) according to the manufacture's guide. In brief, AML-12 cells grown on coverslips were incubated with MitoTracker® Red 200 nM in regular medium for 30 min in a CO₂ incubator at 37 °C. After incubation cells were fixed in ice-cold methanol for 25 min at −20 °C and washed three times with PBS for 5 min. Coverslips were mounted with DAPI and staining was assessed using the BZ-X710 Keyence fluorescence microscope. Ten pictures per coverslip were taken and quantified using ImageJ Software (version 1.52q).

**Mitochondrial membrane potential.** Mitochondrial membrane potential was measured using TMRE (Cat. T669, Thermo) according to the manufacture's guide. In brief, cells were incubated with TMRE 200 nM in regular AML-12 medium for 30 min in a CO₂ incubator at 37 °C. Then cells were washed twice with warmed PBS, fluorescence was measured at Ex/Em: 510/580 nm and normalized to total protein.

**Mitochondrial ROS measurement.** Mitochondrial ROS was measured using the MitoSOX Red kit (M36008 Molecular Probes) according to the manufacture's guide. In brief, cells were incubated with MitoSOX 5 μM in HBSS medium without

FBS for 15 min in a $CO_2$ incubator at 37 °C. Then cells were washed twice with warmed PBS, fluorescence was measured at Ex/Em: 510/580 nm and normalized to total protein.

**ATP level determination**. The levels of ATP were determined using the ATPlite luminescence ATP detection assay system (PerkinElmer) according to the manufacture's guide. Measurement was normalized to total protein.

**Triglycerides measurement**. Triglycerides were measured using the Triglyceride Colorimetric Assay Kit from Cayman (No. 10010303) according to the manufacture's guide. Measurement was normalized to total protein.

**Oil red O staining**. Cells grown on coverslips were washed three times with PBS and fixed with 10% formalin for 1 h at room temperature. Then, cells were washed with Milli-Q water and incubated for 10 min with Oil Red O working solution at room temperature. Solution was removed, cells were washed five times with Milli-Q water and coverslips were mounted. Pictures were taken using the BZ-X710 Keyence microscope.

**Immunofluorescence**. AML-12 cells grown in coverslips were fixed in 4% paraformaldehyde for 15 min at room temperature. Then, cells were washed three times with PBS and permeabilized with PBS containing 0.25% Triton X-100 for 10 min. Cells were washed three times with PBS and then the cells were incubated with blocking buffer (1% Bovine serum Albumin prepared in PBS containing 0.05% Triton X-100). Primary antibodies His-Tag (66005-1-Ig, ProteinTech); TOM70 (14528-1-AP, Proteintech); ATP Synthase β (ab128743); and Golgin-97 (12640-1-AP, Proteintech) were diluted 1:100 in blocking buffer. Cells were incubated with diluted primary antibodies in a humidified container at 4 °C overnight. Cells were then washed in PBS three times and then the cells were incubated with secondary antibodies (1:200 rabbit Alexa fluor 488, Abcam (ab150077), or mouse Cy3, Jackson ImmunoResearch) for 1 h at room temperature. Cells were washed three times in PBS. Nuclei were stained with DAPI and cells were mounted using Prolong mounting medium (Thermo Scientific) before imaging. Images were captured on a Keyence confocal microscope (BZ-X800).

**In vitro kinase assay**. Full-length, human recombinant MATα1 protein (100 ng, Novus Biologicals, Littleton, CO) was incubated with 8.3 ng of active CK2 enzyme (Millipore-Sigma) in the presence of a magnesium/ATP cocktail (Millipore-Sigma) at 0 °C or 30 °C for 1 h. The reaction mixture was immunoprecipitated with 0.5 μg phosphor-Ser antibody and immunoblotted with MATα1 antibody. MATα1 immunoprecipitated without incubation with CK2 served as a -kinase control for the reaction. MATα1 WT and S114 recombinant proteins were generated using the PURExpress® In Vitro Protein Synthesis Kit following manufacturer's instructions and kinase assay was performed as described above.

**CK2 activity**. Casein Kinase II activity was measured in AML-12 cells treated with ethanol ($1 \times 10^6$ cells) using the CycLex CK2 Kinase assay kit (Woburn, MA) according to the manufacturer's recommended protocol.

**MitoPlex targeted quantitative mass spectrometry assay**. A targeted proteomic assay comprising 37 mitochondrial proteins[30] was used on AML-12 cells expressing His-Tag-MATα1 WT or His-Tag-MATα1 S114A treated with ethanol for 48 h (Supplementary data 4). Briefly, cell pellets ($n = 4$ per condition) were lysed in 8 M urea dissolved in 50 mM Tris-HCl buffer, pH 8.0. Thirty-five micrograms of protein from each sample was digested as outlined above using DTT (10 mM), iodoacetamide (100 mM), and a final urea concentration of 2 M. Trypsin was added at a ratio of 1 to 35 μg of total protein and incubate overnight at 37 °C. Digestion was quenched with 1% TFA, and samples were desalted on Nest C18 tips (NestGroup, Southborough, MA). Synthetic stable N[15] isotope-labeled reference peptides derived for targeted endogenous peptides representing the targeted mitochondrial protein were added. Eight duplicates of digested peptides and reference peptide mixture were separated on a Prominence UFLCXR HPLC system (Shimadzu Corp., Kyoto, Japan) with a Waters Xbridge BEH30 C18 2.1 mm × 100 mm, 3.5-μm column flowing at 0.25 ml/min, and 36 °C coupled to a QTRAP 6500 (SCIEX, Framingham, MA). Mobile phase A consisted of 2% acetonitrile, 98% water, and 0.1% formic acid, and mobile phase B consisted of 95% acetonitrile, 5% water, and 0.1% formic acid. After loading, the column was equilibrated with 5% B for 5 min. Peptides were then eluted over 30 min with a linear 5–35% gradient of buffer B. The column was washed with 98% B for 10 min and then returned to 5% B for 5 min before loading the next sample. A scheduled, targeted acquisition method optimized for each peptide was used to monitor fragments within a 2-min window of the optimized retention time. Sample was run in duplicated and only peptides with 40% coefficient of variation (CV) across the heavy standard peptides were used for quantification. Raw data were processed using the Skyline software package (Skyline Daily, version 19.1.1.309) to select peak boundaries and quantify the area under the curve for each fragment monitored. Peptides were manually inspected in Skyline for quality and excluded from further analysis if no discernable peak was observed. The abundance ratios of endogenous to N15 standard peptide

to the endogenous peptide were averaged, and peptide level abundance ratios were further averaged to yield a final abundance ratio for each of the proteins monitored.

**STRING and DAVID analysis**. MATα1 interacting proteins were subjected to Gene Ontology (GO) Cellular Compartments, GO Terms and KEGG pathway enrichment analysis using the STRING (version 11.0) and DAVID (version 6.8) online databases.

**Quantification and statistical analysis**. Data are given as mean ± SEM. Statistical analysis was performed using $t$-tests, ANOVA, and Fisher's tests. All $P$ values were derived from at least three independent experiments. Statistical significance was defined by $p < 0.05$. For mRNA and protein levels, the ratios of genes and proteins to respective housekeeping densitometric values were compared. Calculations were performed using Graphpad (version 9.0.0) and Excel (version 16.54). Densitometry analysis were carried out using ImageJ Software (version 1.52q).

**Reporting summary**. Further information on research design is available in the Nature Research Reporting Summary linked to this article.

## Data availability

All the mass spectrometry proteomics data have been deposited to the ProteomeXchange Consortium via the PRIDE partner repository with the dataset identifier PXD030383. DAVID (https://david.ncifcrf.gov) and STRING (https://string-db.org) are publicly available databases. All other data generated or analyzed during this study are included in this published article (and its Supplementary Information files). Source data are provided with this paper.

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

## Acknowledgements

This work was supported by NIH grants R01AA026759 (S.C.L. and J.E.V.E.), R01DK123763 (S.C.L., J.E.V.E., and J.M.M.), and Plan Nacional of I + D SAF2017-88041-R (J.M.M.). MitoPlex was performed by the Cedars-Sinai Proteomics and Metabolomics Core. The funders had no role in study design, data collection and analysis, decision to publish, or preparation of the manuscript.

## Author contributions

S.C.L. conceived the study. S.C.L. and L.B.T. designed the experiments. Acquisition and data analysis were conducted by L.B.T., B.M., J.W.Y., J.W., M.M., A.R., A.B., W.F., D.F.R., F.L.O., M.L.U., N.M., H.P. and K.R. Human liver specimens were provided by Z.S. and S.L. O.M., J.E.V.E., R.G., E.S., S.L. and J.M.M. provided critical input. Writing and editing of the manuscript were done by L.B.T. and S.C.L. Funding was obtained by S.C.L. and J.E.V.E.

## Competing interests

The authors declare no competing interests.

## Additional information

**Peer review information** *Nature Communications* thanks Kazuhiko Igarashi, Wen-Xing Ding and the other anonymous reviewer(s) for their contribution to the peer review this work. Peer reviewer reports are available.

