## [Peer Review File · Nature Communications]

Reviewers' Comments:

Reviewer #1:

Remarks to the Author:

Alcohol-associated liver disease (ALD) is a leading cause of chronic liver injury worldwide. Earlier work from the author's group demonstrated methionine adenosyltransferase alpha 1 (MATa1), a key enzyme catalyzes the synthesis of S-adenosylmethionine, the principal methyl donor, decreased in ALD. In the present study, using relevant cell culture and mouse ALD models as well as human alcoholic hepatitis (Ah) samples, authors found that mitochondrial MATa1 was selectively depleted in ALD through a mechanism that involves the isomerase PIN1. Alcohol increased MATa1 phosphorylation at Ser114, and this phosphorylation increased its interaction with PIN1 that inhibited its mitochondrial localization. Mass spectrometry analyses showed MATa1 interacted with mitochondrial proteins involved in TCA cycle, oxidative phosphorylation, and fatty acid β -oxidation, which was impaired in human ALD.

The strengths of this manuscript including the use of complementary in vitro and in vivo ALD mouse models as well as human ALD samples, unbiased proteomics and knocking down and overexpression PIN1 as well S114 mutant form of MATa1 in investigating a novel mechanism of MATa1 regulation in ALD. Manuscript was generally well written. The study will advance our current understanding on Mata1 in ALD. However, there were some concerns that need to be addressed to improve the quality of the manuscript.

Specific Comments:

1. Figure 2 E & F, the conclusion would be more convincing if authors can do a reverse IP using MATa1 antibody and then determine the levels of PIN1 bound with MATa1.
2. Can authors comment on why sometimes MATa1 has one band but sometimes has two bands in the same cell line (Figure 3)? Please define "DDK" in the figure legend. The levels of mitochondrial MATa1 should also be determined after ethanol treatment in PIN1 shRNA or overexpression cells (not only showing the basal level changes).
3. Since PIN1 directly binding with MATa1 and negatively regulated MATa1 mitochondrial translocation, there are several important questions that need to be clarified. Is PIN1 a purely cytosolic protein or a portion of PIN1 could be on mitochondria? Where was the interaction between PIN1 and MATa1, cytosol or mitochondria? The levels of phosphorylated MATa1 in cytosol vs mitochondrial should be determined! Answer these questions may help to consolidate the model of which it seemed that PIN1 interacted with phosphorylated S114 MATa1 in the cytosol and sequester MATa1 in the cytosol to prevent its mitochondrial translocation.
4. What is/are the upstream kinases that phosphorylate MATa1? Is an antibody that specifically recognize phosphorylated S114 MATa1 available? This would allow authors to directly assess the changes of this specific phosphorylated MATa1 after alcohol exposure. As the levels of PIN1 did not change by alcohol, thus it seemed that the regulation of phosphorylation of MATa1 is very critical for its interaction and translocation to mitochondria.
5. Figure 6, more data may be needed to support the conclusion that mitochondrial MATa1 protects ethanol-induced mitochondrial dysfunction. Mitotracker Red staining often correlates with mitochondrial mass but not necessarily for mitochondrial functions. A more general screening for more mitochondrial protein changes may be needed to confirm the overall decreased Mitotracker staining. Mitochondrial membrane potential changes and oxygen consumption rate using Seahorse analyzer should be measured in WT and S114A cells with/without ethanol.
6. Figure 7, these interactome MS are very informative and potentially important for the field but some of these key interactions may need to be validated by traditional co-IP experiments.
7. In the discussion part, is it possible that increased mitochondrial MATa1 may increase mitochondrial pool of GSH in addition to altering methylation to protect against ALD? Authors should also discuss the difference of the mouse NIAAA model, in vitro AML12 cells with the more severe human alcoholic hepatitis (AH). As the mouse and cell culture model can only represent the early changes of ALD, these limitations should be discussed in the text. The S114A mutant MATa1 in ALD only tested in vitro cell culture model, which may need to be tested in vivo in the ALD mouse model in the future.

Reviewer #2:

Remarks to the Author:

In this manuscript, Barbier-Torres et al. describe a model whereby PIN1, a prolyl cis/trans isomerase, influences the mitochondrial translocation of MATA1, the enzyme that produces S-adenosylmethionine (SAM). The model is that phosphorylation of MATA1 induces PIN1 binding, which thereby prevents MATA1 translocation to mitochondria. This failed translocation in turn has negative consequences on mitochondrial function. This is relevant to alcohol-associated liver disease (ALD), during which time MATA1 phosphorylation increases, thus diminishing MATA1 levels in mitochondria. Overall, the authors make a reasonably convincing connection between MATA1 phosphorylation, PIN1 interaction, and MATA1 subcellular localization, and the relevance to ALD seems clear. As such, I think there is a strong foundation in place for a Nature Communications publication. However, there are a few missing pieces/controls that seem necessary.

1. Does PIN1 isomerize MATA1, and is that what dictates MATA1 levels? This seems like an obvious and important question.
2. Does phosphorylation of MATA1 affect its stability or activity? The stability and activity of the recombinant S114A mutant should be checked.
3. Does MATA1 produce SAM (and oxidized/reduced glutathione (GSH/GSSG)) in mitochondria, and thus are these levels altered in mitochondria based on MATA1? The authors hypothesize about this in the discussion, but mass spectrometry-based measurements for these molecules are well established.
4. The IP-MS analyses and data reported in Figure 7 and 8 are weak. Many affinity-enrichment mass spectrometry analyses in recent years have shown how uninformative these experiments are when using a single 'bait' protein. It appears that the authors are merely identifying abundant mitochondrial proteins, much like might be seen with a non-specific. (e.g., mito-GFP) control. Furthermore, the diminished interactions of MATA1 in ALD can be driven by the fact that 1) there is less MATA1 in mitochondria, and 2) the expression levels of many mitochondrial proteins is decreased. Finally, the suggestion that MATA1 interaction with all of the proteins is regulatory seems like a major leap. Isn't the more straightforward explanation that MATA1 is affecting SAM and GSH levels? Many mitochondrial pathways rely on SAM (e.g., the assembly of complex I, the production of coenzyme Q, methylation of ribosomal RNA, etc.). Overall, this final stage of the paper either needs to be substantially improved, or eliminated in favor of other more informative (and less speculative) experiments that could shed some light on mechanism (i.e., those listed above).

Minor points

1. I found some figure panels and legends to be a bit unclear (e.g., I don't really follow the experiment in 2D, I don't know what 'DDK' is in Figure 3).
2. Figure 4D is very hard to read and interpret. Also, the actual MS2 spectrum of the peptide validating the phosphorylation site should be reported (supplement is fine). It is important to note that these experiments do not comment on absolute stoichiometry of the phosphorylation site.
3. The legend for figure 4 has two '(D)s' and no '(E).'

Reviewer #3:

Remarks to the Author:

A previous paper from this group showed that methionine adenosyltransferase 1 (MAT1a) is present in mitochondria as well and promotes degradation of cytochrome P450 2E1. However, the function of MAT1a in mitochondria and the mechanism for mitochondria localization have remained elusive. In this manuscript, authors found that MAT1a amount in mitochondria was reduced in the liver of patients with alcoholic liver disease. This was further verified by using a mice model and in vitro treatment of established liver cells with alcohol. As a potential regulator of mitochondrial accumulation, authors identified PIN1, and showed its function by knockdown and a mutant MAT1a. This interaction appeared to regulate half-life of MAT1a in cultured cell. They further found that many mitochondrial proteins interacting with MAT1. These findings are potentially interesting. However, this manuscript lacks experiments that allow mechanistic and biological insights into the function of MAT1a in mitochondria, remaining rather a preliminary report. Importantly, authors did not examine whether the catalytic activity of this enzyme is required in mitochondria or not, and if it is required, how SAM is used there. Without these central issues being addressed, the

manuscript may remain premature.

1. First of all, it is not clear how MAT1a is imported into mitochondria. PIN1 was suggested to inhibit the accumulation in mitochondria in Figs. 1 and 2. However, the other experiments showed that PIN1(phosphorylation) promoted MAT1a degradation. Therefore, PIN1 may just affect the total amount of MAT1a protein, which is not consistent with the results in other figures.
2. Authors found that the amount of MAT1a in mitochondria is reduced in the liver from patients or under disease-mimicking situations. This is a very exciting finding. However, it is not clear which of the findings can explain the mechanism behind. The definitive experiment is to identify the protein kinase for the MAT1a phosphorylation at S114. The findings described are interesting but not connected with each other well.
3. Expression of MAT1a or MAT1a-S114A was carried out in the presence of endogenous MAT1a, which precluded assessment of the contribution of the serine residue in a reasonable manner.
4. The one of the central issues missing from this manuscript is whether SAM synthesis in mitochondria is required for a proper function of this organelle. If so, how SAM is used there. Authors are the best place to answer such questions, and can add much to the literature. Authors should rescue MAT1a knockdown cells with catalytically active or dead MAT1a, with or without S114A mutation to analyze methylation of DNA, RNA or proteins in mitochondria. Authors reported Arg methylation in their previous report. Assessment of mitochondrial function directly dependent on SAM is absolutely required.
5. If S114 is important for the regulation of protein degradation, PIN1 should show some effect on ubiquitination or interaction with ubiquitin E3 adaptor proteins. The connection between S114, PIN1, and protein turnover is not clear in the current form and can be further developed.
6. If MAT1a is phosphorylated upon alcohol stress, why there was no clear band shift in the western blotting? In Fig. 3C there are clearly two bands. However, there is no correlation between the two cell lines whether the upper or the lower band is present in mitochondria.
7. In Fig. 4D, comparison with other phosphorylation sites of MAT1a is needed to establish specific response of S114.
8. To further establish its mitochondrial localization, immunofluorescence staining of wild-type or S114A MAT1a is needed, with several mitochondria proteins and other proteins of other organelle.

Response to reviewers' comments:**Reviewer #1:**

Specific Comments:

1. Figure 2 E & F, the conclusion would be more convincing if authors can do a reverse IP using MATa1 antibody and then determine the levels of PIN1 bound with MATa1.

Response: We thank the reviewer for this suggestion and have done the experiment. As shown in revised Figure 2, we found increased PIN1 levels bound to MAT α 1 in livers of the NIAAA mouse model and AML-12 cells treated with ethanol. For unknown reasons, we haven't been able to perform successfully the reverse IP in human livers.

2. Can authors comment on why sometimes MATa1 has one band but sometimes has two bands in the same cell line (Figure 3)? Please define "DDK" in the figure legend. The levels of mitochondrial MATa1 should also be determined after ethanol treatment in PIN1 shRNA or overexpression cells (not only showing the basal level changes).

Response: We speculate the main reason is that in some of the blots the gel was not resolved enough. Since MAT α 1's molecular weight is 48 KDa, we speculate that the higher molecular weight band may represent a post-translationally modified form of MAT α 1.

We have defined DDK in the figure legend as suggested.

We have evaluated mitochondrial MAT α 1 levels after ethanol treatment in *Pin1* silenced AML-12 cells and the results show that PIN1 interferes with MAT α 1 mitochondrial targeting in ALD as its mitochondrial content is significantly higher in *siPin1* cells. New data has been added to revised Figure 3. We believe that the experiment overexpressing PIN1 is not as relevant as ethanol itself depletes mitochondrial MAT α 1 and PIN1 overexpression also reduces it at baseline.

3. Since PIN1 directly binding with MATa1 and negatively regulated MATa1 mitochondrial translocation, there are several important questions that need to be clarified. Is PIN1 a purely cytosolic protein or a portion of PIN1 could be on mitochondria? Where was the interaction between PIN1 and MATa1, cytosol or mitochondria? The levels of phosphorylated MATa1 in cytosol vs mitochondrial should be determined! Answer these questions may help to consolidate the model of which it seemed that PIN1 interacted with phosphorylated S114 MATa1 in the cytosol and sequester MATa1 in the cytosol to prevent its mitochondrial translocation.

Response: We thank the reviewer for these excellent points. Although PIN1 has been detected in the mitochondrial outer membrane in neurons, we did not detect any PIN1 in our mitochondrial fractions (Figures 3c and 3d, and revised Figures S1f and S1g). Hence, we evaluated PIN1 and MAT α 1 interaction in the cytosol of ethanol-fed livers and found it increased in ALD. This data is included in revised Figure 2h.

For MAT α 1 phosphorylation we ran both cytosolic and mitochondrial fractions from AML-12 cells treated with ethanol in Phos-tag™ gels that allow separation of the phosphorylated forms from the unphosphorylated form of the same protein. As shown in revised Figure 4b, we found that ethanol induced MAT α 1 phosphorylation in the cytosol while its phosphorylation within mitochondria remained unaltered relative to its total content. Taken together, our results support that in ALD MAT α 1 is hyperphosphorylated at Ser114 in the cytosol, which enhances its interaction with PIN1 and prevents its mitochondrial translocation.

4. What is/are the upstream kinases that phosphorylate MAT α 1? Is an antibody that specifically recognize phosphorylated S114 MAT α 1 available? This would allow authors to directly assess the changes of this specific phosphorylated MAT α 1 after alcohol exposure. As the levels of PIN1 did not change by alcohol, thus it seemed that the regulation of phosphorylation of MAT α 1 is very critical for its interaction and translocation to mitochondria.

Response: These are excellent questions and we have done extensive investigations to identify the upstream kinase. MAT α 1 phosphorylation at Ser114 has not been reported before and there is no antibody that recognizes phospho-MAT α 1-S114 available. Based on online kinase predictions, we investigated a number of kinases. We excluded JNK, GSK β , and p38 kinase because they were either unchanged or not consistently changed in the three models of ALD (human ALD, NIAAA model, and AML-12 cells treated with ethanol) (see Figure S4b-c). CK2 is also a kinase on the list but its role in ALD has not been reported to our knowledge. We found total CK2 content is higher in all three ALD models and its activity increased in AML-12 cells treated with ethanol. In vitro kinase assays showed that CK2 phosphorylates MAT α 1, and using recombinant MAT α 1 WT and S114A we demonstrated that CK2 phosphorylates it at Ser114. Moreover, we found that *Csnk2a1* silencing prevented ethanol-induced depletion of MAT α 1 mitochondrial content and thereby protected against ethanol-induced mitochondrial dysfunction and accumulation of triglycerides in AML-12 cells. This new data has been added to new Figure 7. Altogether, we found that both ethanol-induced MAT α 1 phosphorylation at Ser114 by CK2 and subsequent binding to PIN1 are critical events that prevent MAT α 1 from translocating into mitochondria.

5. Figure 6, more data may be needed to support the conclusion that mitochondrial MAT α 1 protects ethanol-induced mitochondrial dysfunction. Mitotracker Red staining often correlates with mitochondrial mass but not necessarily for mitochondrial functions. A more general screening for more mitochondrial protein changes may be needed to confirm the overall decreased Mitotracker staining. Mitochondrial membrane potential changes and oxygen consumption rate using Seahorse analyzer should be measured in WT and S114A cells with/without ethanol.

Response: We have done as suggested by the reviewer. Mitochondrial respiration was evaluated using Seahorse analyzer and membrane potential was measured using TMRE dye. This new data has been added to revised Figure 6.

6. Figure 7, these interactome MS are very informative and potentially important for the field but some of these key interactions may need to be validated by traditional co-IP experiments.

Response: As suggested by the reviewer we have validated the interaction between MAT α 1 and some of the mitochondrial proteins identified by MS in human liver by traditional co-IP. We found interaction of MAT α 1 with SDHA, CPT1 and ACADM. These results have been added to revised Figure 8b.

7. In the discussion part, is it possible that increased mitochondrial MAT α 1 may increase mitochondrial pool of GSH in addition to altering methylation to protect against ALD? Authors should also discuss the difference of the mouse NIAAA model, in vitro AML12 cells with the more severe human alcoholic hepatitis (AH). As the mouse and cell culture model can only represent the early changes of ALD, these limitations should be discussed in the text. The S114A mutant MAT α 1 in ALD only tested in vitro cell culture model, which may need to be tested in vivo in the ALD mouse model in the future.

Response: We thank the reviewer for these comments and have added them to the discussion. We tried to measure SAME and GSH levels in isolated mitochondria from AML-12 cells expressing WT or S114A mutant of MAT α 1 \pm ethanol multiple times both in Los Angeles and Spain using different approaches but found too much variability between experiments to trust the values. We are not clear what led to the high variability, but suspect there may be “leakage” of metabolites during sample preparation, which involves suspending the mitochondria in different buffers. These attempts have taken more than three months. In its place, we have measured levels of methylated protein as an alternate index of the methylation capacity within the mitochondria. This new result is shown in revised Figure 8e, which shows ethanol lowers methylated proteins in empty vector and WT MAT1A expressing AML12 cells but this is attenuated in S114A expressing cells.

Besides being the main methyl donor in the cell, SAME is also a precursor of GSH. However, previous studies have shown that GSH is not produced within mitochondria but produced in the cytosol and transported into the organelle, which suggests that mitochondrial SAME's protective effect might be mainly via methylation of variety of substrates such as RNA, DNA and proteins. Unfortunately, as stated above we were not able to measure mitochondrial GSH levels reliably.

Our study used two experimental models and two human ALD stages. The NIAAA model consists of a chronic-plus-single-binge feeding and induces liver injury, inflammation and fatty liver, mimicking acute-on-chronic alcoholic liver injury in patients. The model of AML-12 cells treated with ethanol recapitulates only the early stage of ALD but allows molecular dissection of the effect of ethanol on hepatocytes and its organelles. Key changes from these models were confirmed using human AH and end-stage cirrhotic ALD samples, which are limited in quantity. Using this approach, we

found mitochondrial MAT α 1 is selectively reduced in ALD, as seen in human AH liver samples, NIAAA model and AML-12 cells treated with ethanol. More importantly, preserving mitochondrial MAT α 1 protected against alcohol-induced mitochondrial injury and triglyceride accumulation, suggesting that mitochondrial MAT α 1 depletion is important in ALD pathogenesis. In summary, we show that mitochondrial MAT α 1 is important for hepatic mitochondrial function and that its depletion occurs early and persists to end-stage ALD through a mechanism that involves its phosphorylation at Ser114 by CK2 and interaction with PIN1.

We have added this to the discussion.

Reviewer #2:

1. Does PIN1 isomerize MAT α 1, and is that what dictates MAT α 1 levels? This seems like an obvious and important question.

Response: This is a very important question. To assess whether PIN1 isomerizes MAT α 1 and that dictates its mitochondrial translocation, we overexpressed PIN1 WT and the PIN1 catalytic mutant R68A/R69A, and evaluated MAT α 1 total and mitochondrial levels. R68/R69 are in the prolyl isomerase flexible loop of PIN1 and their mutation shows reduced isomerase activity by >500-fold (Yaffe et al., 1997). As shown in revised Figure 3I, while PIN1 WT significantly reduced MAT α 1 mitochondrial content, mitochondrial MAT α 1 level remain unchanged when PIN1 mutant was overexpressed. These findings prove that MAT α 1 isomerization by PIN1 is required to inhibit MAT α 1 mitochondrial targeting in ALD.

2. Does phosphorylation of MAT α 1 affect its stability or activity? The stability and activity of the recombinant S114A mutant should be checked.

Response: We agree with the reviewer that this is an important question. Our results show that phosphorylation at Ser114 regulates MAT α 1's stability. As shown in Figure 5e, while ethanol-induced phosphorylation reduced MAT α 1's half-life by more than 50%, MAT α 1 S114A mutant remained stable. As expected, we found that the stability at baseline is comparable between both as phosphorylation at that specific residue might be very low. For MAT activity, recombinant MAT α 1 WT and S114A were generated and SAME production and L-methionine consumption were measured. We found that MAT α 1 S114A mutant has less activity as compared to WT but it is active and able to produce SAME. This new data has been added to revised Figure 5a.

3. Does MAT α 1 produce SAM (and oxidized/reduced glutathione (GSH/GSSG)) in mitochondria, and thus are these levels altered in mitochondria based on MAT α 1? The authors hypothesize about this in the discussion, but mass spectrometry-based measurements for these molecules are well established.

Response: This is a very important question. We tried to measure SAME and GSH levels in isolated mitochondria from AML-12 cells expressing WT or S114A mutant of MAT α 1 \pm ethanol multiple times in Los Angeles and Spain using different approaches

but found too much variability between experiments to trust the values. We are not clear what led to the high variability, but suspect there may be “leakage” of metabolites during sample preparation, which involves suspending the mitochondria in different buffers. Ethanol is known to damage mitochondria, so this likely contributed to the high variability. These attempts have taken more than three months. In its place, we have measured levels of methylated protein as an alternate index of the methylation capacity within the mitochondria. This new result is shown in revised Figure 8e, which shows ethanol lowers methylated proteins in empty vector and WT MAT1A expressing AML12 cells but this is attenuated in S114A expressing cells.

Please see response to Review one, under comment #7 regarding GSH/GSSG levels.

4. The IP-MS analyses and data reported in Figure 7 and 8 are weak. Many affinity-enrichment mass spectrometry analyses in recent years have shown how uninformative these experiments are when using a single ‘bait’ protein. It appears that the authors are merely identifying abundant mitochondrial proteins, much like might be seen with a non-specific. (e.g., mito-GFP) control. Furthermore, the diminished interactions of MAT α 1 in ALD can be driven by the fact that 1) there is less MAT α 1 in mitochondria, and 2) the expression levels of many mitochondrial proteins is decreased. Finally, the suggestion that MAT α 1 interaction with all of the proteins is regulatory seems like a major leap. Isn't the more straightforward explanation that MAT α 1 is affecting SAM and GSH levels? Many mitochondrial pathways rely on SAM (e.g., the assembly of complex I, the production of coenzyme Q, methylation of ribosomal RNA, etc.). Overall, this final stage of the paper either needs to be substantially improved, or eliminated in favor of other more informative (and less speculative) experiments that could shed some light on mechanism (i.e., those listed above).

Response: We agree with the reviewer and have revised in this part. First, we have validated the interaction between MAT α 1 and some mitochondrial proteins (SDHA, CPT1 and ACADM). Although the expression of some mitochondrial proteins is reduced in ALD, we found that many mitochondrial proteins were preserved after ethanol treatment in AML-12 cells expressing MAT α 1 S114A mutant, which suggest that the presence of MAT α 1 within mitochondria positively regulates these proteins' abundance (see Figure 8c-d) along with mitochondrial function. The severity of the disease in humans is more advanced as compared to our in vitro model and altogether, our findings suggest that mitochondrial MAT α 1 is playing a major role in maintaining mitochondrial integrity and function. Our hypothesis is since SAME has a short half-life and is needed by numerous substrates, having MAT α 1 in close proximity ensures a source of SAME for its key substrates. In fact, there was a paper published during this revision that studied the role of mitochondrial SAME (Schober *et al.*, 2021). Similar to our findings, the authors show that mitochondrial SAME is required for OXPHOS and TCA cycle metabolism. They found a profound mitochondrial defect affecting both mitochondrial translation and OXPHOS with both nuclear and mitochondrial encoded subunits of complexes I, III and IV severely decreased. It should be noted that this study concludes that SAMC (SAME channel) is the only source of SAME in mitochondria but none of the studies were performed in cells that express MAT1A. Our results

demonstrate that MAT α 1 is another source of mitochondrial SAME in the liver and that higher levels of mitochondrial MAT α 1 help preserve mitochondrial proteins and function via SAME production.

We have added these to the discussion.

Minor points:

1. I found some figure panels and legends to be a bit unclear (e.g., I don't really follow the experiment in 2D, I don't know what 'DDK' is in Figure 3).

Response: We apologize and have revised figure legends to improve clarity.

2. Figure 4D is very hard to read and interpret. Also, the actual MS2 spectrum of the peptide validating the phosphorylation site should be reported (supplement is fine). It is important to note that these experiments do not comment on absolute stoichiometry of the phosphorylation site.

Response: We thank the reviewer for bringing the absolute stoichiometry of the phosphorylation site to our attention. We have added a sentence (see below) in the discussion but determining absolute stoichiometry of this phospho-site is outside the scope of this manuscript and we plan on addressing this point in future work.

"Although phosphorylation of MAT α 1 Ser114 is altered, these experiments do not allow us to determine the stoichiometry of this phospho-site, which may play a key role in the regulation of MAT α 1 interaction with PIN1 and mitochondrial targeting".

At the reviewer's request, we have added a representative spectra obtained from DDA-MS acquisitions MAT α 1 immunoprecipitations in the supplement (Figure S2d) showing all of the ions used to identify the peptide containing MAT α 1 S114. Many of these ions contain neutral losses indicative of phosphorylation of MAT α 1 Ser114. The methodology for these acquisitions was also included in the methods section.

3. The legend for figure 4 has two '(D)s' and no '(E).'

Response: We thank the reviewer for this and have corrected it.

Reviewer #3:

A previous paper from this group showed that methionine adenosyltransferase 1 (MAT1a) is present in mitochondria as well and promotes degradation of cytochrome P450 2E1. However, the function of MAT1a in mitochondria and the mechanism for mitochondria localization have remained elusive. In this manuscript, authors found that MAT1a amount in mitochondria was reduced in the liver of patients with alcoholic liver disease. This was further verified by using a mice model and in vitro treatment of established liver cells with alcohol. As a potential regulator of mitochondrial accumulation, authors identified PIN1, and showed its function by knockdown and a

mutant MAT1a. This interaction appeared to regulate half-life of MAT1a in cultured cell. They further found that many mitochondrial proteins interacting with MAT1. These findings are potentially interesting. However, this manuscript lacks experiments that allow mechanistic and biological insights into the function of MAT1a in mitochondria, remaining rather a preliminary report. Importantly, authors did not examine whether the catalytic activity of this enzyme is required in mitochondria or not, and if it is required, how SAM is used there. Without these central issues being addressed, the manuscript may remain premature.

1. First of all, it is not clear how MAT1a is imported into mitochondria. PIN1 was suggested to inhibit the accumulation in mitochondria in Figs. 1 and 2. However, the other experiments showed that PIN1 (phosphorylation) promoted MAT1a degradation. Therefore, PIN1 may just affect the total amount of MAT1a protein, which is not consistent with the results in other figures.

Response: Although MAT α 1 does not have a mitochondrial targeting sequence it has a TOM20 recognition motif (aa79-83). The exact mechanism by which MAT α 1 normally translocates into mitochondria remains to be studied and we believe is out of the scope of this work.

Regarding MAT α 1 regulation by PIN1, we have shown that PIN1 does not alter MAT α 1 total expression but its mitochondrial localization at basal conditions (Figure 3a-d). In the presence of ethanol, we have new data showing that total MAT α 1 protein level falls by increased degradation via autophagy (Figure S2b-c). However, ethanol also specifically impairs MAT α 1 mitochondrial targeting via CK2 and PIN1.

2. Authors found that the amount of MAT1a in mitochondria is reduced in the liver from patients or under disease-mimicking situations. This is a very exiting finding. However, it is not clear which of the findings can explain the mechanism behind. The definitive experiment is to identify the protein kinase for the MAT1a phosphorylation at S114. The findings described are interesting but not connected with each other well.

Response: We agree with the reviewer that identifying the upstream kinase is important and in the revision have done so. Based on online kinase predictions, we investigated a number of kinases. We excluded JNK, GSK β , and p38 kinase because they were either unchanged or not consistently changed in the three models of ALD (human ALD, NIAAA model, and AML-12 cells treated with ethanol) (see Figure S4b-c). CK2 is also a kinase on the list but its role in ALD has not been reported. We found total CK2 content is higher in all three ALD models and its activity increased in AML-12 cells treated with ethanol. In vitro kinase assays showed that CK2 phosphorylates MAT α 1, and using recombinant MAT α 1 WT and S114A we demonstrated that CK2 phosphorylates it at Ser114. Moreover, we found that *Csnk2a1* silencing prevented ethanol-induced depletion of MAT α 1 mitochondrial content and thereby protected against ethanol-induced mitochondrial dysfunction and accumulation of triglycerides in AML-12 cells. This new data has been added to new Figure 7. Altogether, we found that both ethanol-

induced MAT α 1 phosphorylation at Ser114 by CK2 and subsequent binding to PIN1 are critical events that prevent MAT α 1 from translocating into mitochondria.

3. Expression of MAT1a or MAT1a-S114A was carried out in the presence of endogenous MAT1a, which precluded assessment of the contribution of the serine residue in a reasonable manner.

Response: We have silenced endogenous *Mat1a* in AML-12 cells using a specific siRNA and have evaluated mitochondrial function by measuring mROS and ATP levels and measured triglyceride accumulation after MAT α 1 WT and S114A overexpression and ethanol treatment. As shown in new Figure S3, the same protective effect with MAT α 1 S114A was observed. It is important to note that we also silenced *Pin1* and *Csnk2a1*, which allowed us to study the contribution of Ser114 residue on endogenous MAT α 1. By inhibiting the kinase that phosphorylates MAT α 1 at Ser114 and the isomerase PIN1 that specifically binds MAT α 1 at Ser114 we confirmed that the contribution of that specific residue is key for MAT α 1 mitochondrial localization.

4. The one of the central issues missing from this manuscript is whether SAM synthesis in mitochondria is required for a proper function of this organelle. If so, how SAM is used there. Authors are the best place to answer such questions, and can add much to the literature. Authors should rescue MAT1a knockdown cells with catalytically active or dead MAT1a, with or without S114A mutation to analyze methylation of DNA, RNA or proteins in mitochondria. Authors reported Arg methylation in their previous report. Assessment of mitochondrial function directly dependent on SAM is absolutely required.

Response: We tried to measure SAME levels in isolated mitochondria from AML-12 cells expressing WT or S114A mutant of MAT α 1 \pm ethanol multiple times in Los Angeles and Spain using different approaches but found too much variability between experiments to trust the values. We are not clear what led to the high variability, but suspect there may be “leakage” of metabolites during sample preparation, which involves suspending the mitochondria in different buffers. Ethanol is known to damage mitochondria, so this likely contributed to the high variability. These attempts have taken more than three months. In its place, we have measured levels of methylated protein as an alternate index of the methylation capacity within the mitochondria. This new result is shown in revised Figure 8e, which shows ethanol lowers methylated proteins in empty vector and WT MAT1A expressing AML12 cells but this is attenuated in S114A expressing cells.

There was a paper published during this revision that studied the role of mitochondrial SAME (Schober *et al.*, 2021). Similar to our findings, the authors show that mitochondrial SAME is required for OXPHOS and TCA cycle metabolism. They found a profound mitochondrial defect affecting both mitochondrial translation and OXPHOS with both nuclear and mitochondrial encoded subunits of complexes I, III and IV severely decreased. It should be noted that this study concludes that SAMC (SAME channel) is the only source of SAME in mitochondria but none of the studies were performed in cells that express MAT1A. Our results demonstrate that MAT α 1 is another

source of mitochondrial SAME in the liver and that higher levels of mitochondrial MAT α 1 help preserve mitochondrial proteins and function via SAME production. Moreover, our new results showing that both *Pin1* and *Csnk2a1* silencing preserve endogenous MAT α 1 mitochondrial content thereby improving mitochondrial function after ethanol treatment further support our hypothesis. Altogether, our results provide evidence that mitochondrial MAT α 1 is important to maintain mitochondrial SAME levels and mitochondrial function.

5. If S114 is important for the regulation of protein degradation, PIN1 should show some effect on ubiquitination or interaction with ubiquitin E3 adaptor proteins. The connection between S114, PIN1, and protein turnover is not clear in the current form and can be further developed.

Response: We thank the reviewer for raising this important question. Although PIN1 can induce the proteasomal degradation of target proteins, we found that it is not the mechanism by which ethanol reduces MAT α 1. Ethanol is known to inhibit proteasomal activity and we found that it led to the accumulation of ubiquitinated proteins while having a negative effect on MAT α 1 in AML-12 cells, suggesting that ethanol induces MAT α 1 degradation through another mechanism. Moreover, treatment with the proteasome inhibitor MG132 had no effect on MAT α 1 protein levels. On the other hand, ethanol is known to activate autophagy and we found that autophagy inhibition using chloroquine blocked MAT α 1 degradation, which suggests that MAT α 1 is regulated by autophagy in ALD. Altogether, these results support the hypothesis that along with impaired mitochondrial targeting, ethanol promotes MAT α 1 lysosomal degradation. These new data has been added to revised Figure S2a-c.

6. If MAT1a is phosphorylated upon alcohol stress, why there was no clear band shift in the western blotting? In Fig. 3C there are clearly two bands. However, there is no correlation between the two cell lines whether the upper or the lower band is present in mitochondria.

Response: There are many proteins that do not show a clear band shift upon phosphorylation, including MAT α 2 (Ramani K. et al., 2014). Phosphorylation causes motility shift in the rare instances where it causes a change in tertiary structure that affects motility even after denaturation. Regarding the two bands, we speculate the main reason is that in some of the blots the gel was not resolved enough. Since MAT α 1's molecular weight is 48 KDa, we speculate that the higher molecular weight band may represent a post-translationally modified form of MAT α 1.

7. In Fig. 4D, comparison with other phosphorylation sites of MAT1a is needed to establish specific response of S114.

Response: There are three other MAT α 1 phosphorylation sites that have been reported (S73, Y296, T341) (<https://www.phosphosite.org/proteinAction.action?id=5383&showAllSites=true>). However, we did not detect any of them by MS. Regarding the specific response of

S114, we consider that the mutation of that residue to an alanine and overexpression of both MAT α 1 WT and S114A in AML-12 for comparison establishes a very specific response. In addition, we found that CK2 is the kinase responsible for the phosphorylation of MAT α 1 at Ser114 and that *Csnk2a1* silencing prevents ethanol-induced MAT α 1 mitochondrial depletion and mitochondrial injury. Similarly, we found that *Pin1* silencing (MAT α 1 Ser114-Pro115 is the only PIN1 binding motif) also prevents ethanol-induced MAT α 1 mitochondrial depletion. Altogether, these findings strongly support that MAT α 1 phosphorylation at Ser114 is key for its interaction with PIN1 and mitochondrial depletion in ALD.

8. To further establish its mitochondrial localization, immunofluorescence staining of wild-type or S114A MAT1a is needed, with several mitochondria proteins and other proteins of other organelle.

Response: We have added immunofluorescence staining using His-Tag, TOM70 (mitochondrial outer membrane), ATPB (mitochondrial matrix) and Golgin-97 (golgi apparatus) antibodies as suggested. This new data has been added to revised Figure 5h and new Figure S3a-b and strongly support our findings that ethanol specifically reduces mitochondrial MAT α 1.

Reviewers' Comments:

Reviewer #1:

Remarks to the Author:

Authors have done an excellent job and performed additional experiments to address my concerns. My concerns have been adequately addressed and the revised manuscript will advance our understanding on mitochondria MAT1a in alcoholic liver disease.

Reviewer #2:

Remarks to the Author:

I continue to find that the authors have shown an interesting, yet murky, connection between MATa1 phosphorylation, PIN1 interaction, and MATa1 subcellular localization, with relevance to ALD. However, despite some welcome additions (e.g., the PIN1 mutant data in Fig. 3l), I'm concerned that the new data presented by the authors do not support their model. For example:

1. The new Phos-Tag gels in Fig. 2b show that the overwhelming amount of MATa1 is non-phosphorylated, and thus should be free to translocate to mitochondria (i.e., it would not be prevented by the phosphorylation-dependent PIN1 binding).
2. The ratio of phosphorylated to non-phosphorylated MATa1 in mitochondria is higher than in the cytosol (same panel), which is opposite of what the model proposes.
3. In Figure 7g, it's clear that the silencing of CK2a leads to more MATa1 overall, but not more MAT1a in mitochondria. That is, in the absence of the supposed kinase that would prevent mitochondrial localization of MAT1a, a lower percentage of total MAT1a is mitochondrial.
4. The new Seahorse data in Fig. 7i does not appear to be properly normalized, as the oligomycin-treated control and silenced cells have different OCR levels. While that is possible, it would mean the control cells are more uncoupled. Either way, these data do not support a strong effect of CK2a silencing on mitochondrial dysfunction.

Other data remains indirect or speculative. For example:

1. It is very hard to draw any clear and direct conclusions from the general protein methylation blots. It's unfortunate that the authors couldn't make the more relevant SAM measurement, especially given that others have done so (e.g., via mito-tagged cells).
2. It also remains very difficult to draw any solid conclusions from the protein-protein interaction data. I stand by my original comments on this. The suggestion that MATa1 interaction with all of these abundant mitochondrial proteins is regulatory is a major leap. The authors state that, "Our hypothesis is since SAME has a short half-life and is needed by numerous substrates, having MATa1 in close proximity ensures a source of SAME for its key substrates." I don't follow this logic, as most, if not all, of the identified interacting proteins do not use SAM.
3. Unequivocally identifying the relevant kinase is extremely difficult and I don't think it needs to be part of this story. Suggesting CK2a is fine, based on its interesting induction and in vitro capabilities. However, most kinases would probably phosphorylate MATa1 in vitro. I'm not convinced of the direct connection here (in part based on what I mention above), and thus claims should be calibrated appropriately.

Reviewer #3:

Remarks to the Author:

Authors have added several new experiments to address comments from this reviewer. Most of the results are strongly supporting their original contentions. Especially, identification of CK2 as a kinase of Mat1 regulation is critically important. The alterations in methylation of mitochondrial proteins are also supporting the conclusions. (In the future, identification of these methylated proteins will be very interesting!) The new results in Fig. 5h showing mitochondrial localization appear not conclusive, but other results are consistent with its nuclear function. Congratulations on this beautiful interesting work.

REVIEWER COMMENTS

Reviewer #1 (Remarks to the Author):

Authors have done an excellent job and performed additional experiments to address my concerns. My concerns have been adequately addressed and the revised manuscript will advance our understanding on mitochondria MAT1a in alcoholic liver disease.

We thank the reviewer for the nice comments.

Reviewer #2 (Remarks to the Author):

We thank the reviewer for the constructive comments. We would like to emphasize that although we have found a mechanism by which MAT α 1 mitochondrial localization is impaired in ALD, this might not be the only mechanism by which MAT α 1 mitochondrial targeting and stability are regulated. It is noteworthy that our group has only reported MAT α 1's mitochondrial presence very recently and this is a new area of study. In fact, how MAT α 1 is naturally imported into mitochondria remains to be elucidated.

I continue to find that the authors have shown an interesting, yet murky, connection between MAT α 1 phosphorylation, PIN1 interaction, and MAT α 1 subcellular localization, with relevance to ALD. However, despite some welcome additions (e.g., the PIN1 mutant data in Fig. 3I), I'm concerned that the new data presented by the authors do not support their model. For example:

1. The new Phos-Tag gels in Fig. 2b show that the overwhelming amount of MAT α 1 is non-phosphorylated, and thus should be free to translocate to mitochondria (i.e., it would not be prevented by the phosphorylation-dependent PIN1 binding).

Response: As stated above, how MAT α 1 is naturally imported into mitochondria remains to be elucidated. The Phos-Tag gels confirm that ethanol increased the phosphorylation of a fraction of cytosolic MAT α 1, which we speculate cannot translocate into mitochondria after binding to PIN1. It is important to highlight that MAT α 1, a fraction of which is phosphorylated at baseline, is mainly cytosolic. Hence, the assumption that non-phosphorylated MAT α 1, which is most of MAT α 1, is free to translocate to mitochondria is not correct. If that assumption were true, majority of MAT α 1 would be mitochondrial.

2. The ratio of phosphorylated to non-phosphorylated MAT α 1 in mitochondria is higher than in the cytosol (same panel), which is opposite of what the model proposes.

Response: We respectfully disagree with the reviewer that this result is opposite to what we propose. Part of this may be due to lack of clarity on our part. As discussed above, we found that a proportion of cytosolic MAT α 1 is hyperphosphorylated by alcohol at S114, which we speculate cannot translocate into mitochondria after binding to PIN1. We have clearly shown that S114A mutation prevented MAT α 1 interaction with PIN1 and its mitochondrial content was not impacted by ethanol. Importantly, Phos-Tag

gels capture phosphorylated Ser/Thr/Tyr and His/Asp/Lys. It is possible that MAT α 1 is phosphorylated at these other residues but the proteomics method we used mainly picks up pSer and pThr. A different approach would be needed to pick up pTyr and the acidic conditions during sample preparation for our proteomics method would destroy pHis. In contrast to S114, phosphorylation at other residues may not affect MAT α 1's mitochondrial targeting. Clearly we are only at the beginning of understanding how mitochondrial MAT α 1 targeting is regulated or dysregulated and much more work is needed to unravel how MAT α 1 phosphorylation at different residues affects its subcellular localization and functionality.

3. in Figure 7g, it's clear that the silencing of CK2a leads to more MATa1 overall, but not more MAT1a in mitochondria. That is, in the absence of the supposed kinase that would prevent mitochondrial localization of MAT1a, a lower percentage of total MAT1a is mitochondrial.

Response: We apologize for this confusion, as the densitometric data of total MAT α 1 was not included in Figure 7h previously. We have now added that data in Fig. 7h and knocking down CK2a did not change the total MAT α 1 level but it protected against ethanol-induced depletion (both total and in the mitochondria). It is possible that at baseline there is minimal to no CK2a-mediated S114 phosphorylation and that's why knocking down CK2a had no effect on mitochondrial MAT α 1 content in the absence of ethanol.

4. The new Seahorse data in Fig. 7i does not appear to be properly normalized, as the oligomycin-treated control and silenced cells have different OCR levels. While that is possible, it would mean the control cells are more uncoupled. Either way, these data do not support a strong effect of CK2a silencing on mitochondrial dysfunction.

Response: We thank the reviewer for pointing this out as it made us realize that the OCR units were wrong. All Seahorse experiments were normalized to protein concentration so the unit should have been pmol/min/ μ g protein. We also caught an error in the OCR calculation. These have been corrected in the revised figures. In order to study mitochondrial respiration, all groups were treated with the ATP synthase inhibitor oligomycin. We evaluated the protective effect of CK2a silencing on mitochondrial respiration (basal respiration, ATP production and maximal respiratory capacity) after ethanol treatment but had not calculated the proton leak between control and silenced groups for coupling efficiency. Following the reviewer's comment, we calculated the proton leak and found a small but not significant reduction after *Csnk2a1* silencing. This, along with slightly higher levels of ATP-linked OCR, suggests that CK2a could have a negative impact on the coupling of the electron transport chain. CK2a has numerous targets and participates in many different biological pathways so it is possible that it alters hepatic mitochondrial metabolism. Nevertheless, CK2a silencing clearly prevented mitochondrial MAT α 1 depletion and protected against ethanol-induced mitochondrial dysfunction and triglyceride accumulation. We believe that a more modest effect of CK2a inhibition is not surprising considering silencing is never complete and you only need a tiny amount of kinase to phosphorylate multiple targets.

Other data remains indirect or speculative. For example:

1. It is very hard to draw any clear and direct conclusions from the general protein methylation blots. It's unfortunate that the authors couldn't make the more relevant SAM measurement, especially given that others have done so (e.g., via mito-tagged cells).

Response: We thank the reviewer for this suggestion, as we were not familiar with it. Although this technique is described to be used for the rapid and specific isolation of intact mitochondria, it has not been used in ethanol-treated cells. It is well known that ethanol causes mitochondrial damage and this has been the limitation for mitochondrial SAME measurement in our study.

2. It also remains very difficult to draw any solid conclusions from the protein-protein interaction data. I stand by my original comments on this. The suggestion that MAT α 1 interaction with all of these abundant mitochondrial proteins is regulatory is a major leap. The authors state that, "Our hypothesis is since SAME has a short half-life and is needed by numerous substrates, having MAT α 1 in close proximity ensures a source of SAME for its key substrates." I don't follow this logic, as most, if not all, of the identified interacting proteins do not use SAM.

Response: We apologize that the sentence may have been confusing. Our hypothesis is that MAT α 1 interacts with these proteins to ensure a source of SAME for their own methylation. None of the proteins identified in this work are methyl-transferases and we never meant that they used SAME as a substrate. Supporting our findings and included in the discussion, Schober et al., published recently the importance of SAME for mitochondrial protein methylation and included a list of mitochondrial proteins found to be methylated. We have revised the discussion to clarify this. We have also added known methylation sites of the mitochondrial MAT α 1-interacting proteins. We found at least 69% of these interacting proteins have known methylation sites. This information has been added to Supplemental Table I.

3. Unequivocally identifying the relevant kinase is extremely difficult and I don't think it needs to be part of this story. Suggesting CK2a is fine, based on its interesting induction and in vitro capabilities. However, most kinases would probably phosphorylate MAT α 1 in vitro. I'm not convinced of the direct connection here (in part based on what I mention above), and thus claims should be calibrated appropriately.

Response: Little is known about CK2 in ALD and we believe that this piece of data is important and valued by the other two reviewers. CK2 was able to phosphorylate MAT α 1 WT but not MAT α 1 S114A mutant, confirming that CK2 phosphorylates MAT α 1 at Ser114 and this is not a non-specific event. Additionally, its induction and increased activity in ALD and the prevention of mitochondrial MAT α 1 depletion after its silencing confirmed that CK2 is involved in lowering MAT α 1 mitochondrial levels in ALD.

Reviewer #3 (Remarks to the Author):

Authors have added several new experiments to address comments from this reviewer. Most of the results are strongly supporting their original contentions. Especially, identification of CK2 as a kinase of Mat1 regulation is critically important. The alterations

in methylation of mitochondrial proteins are also supporting the conclusions. (In the future, identification of these methylated proteins will be very interesting!) The new results in Fig. 5h showing mitochondrial localization appear not conclusive, but other results are consistent with its nuclear function. Congratulations on this beautiful interesting work.

We thank the reviewer for the nice comments.

Reviewers' Comments:

Reviewer #2:

Remarks to the Author:

While I continue to find some aspects of the manuscript to be overstated, the authors have made clear and valuable contributions to the field. I have no further critiques.